# Nutrient status shapes selfish mitochondrial genome dynamics across different levels of selection

Bryan L Gitschlag[1], Ann T Tate[1], Maulik R Patel[1,2,3]*

[1]Department of Biological Sciences, Vanderbilt University, Nashville, United States; [2]Department of Cell and Developmental Biology, Vanderbilt University School of Medicine, Nashville, United States; [3]Diabetes Research and Training Center, Vanderbilt University School of Medicine, Nashville, United States

**Abstract** Cooperation and cheating are widespread evolutionary strategies. While cheating confers an advantage to individual entities within a group, competition between groups favors cooperation. Selfish or cheater mitochondrial DNA (mtDNA) proliferates within hosts while being selected against at the level of host fitness. How does environment shape cheater dynamics across different selection levels? Focusing on food availability, we address this question using heteroplasmic *Caenorhabditis elegans*. We find that the proliferation of selfish mtDNA within hosts depends on nutrient status stimulating mtDNA biogenesis in the developing germline. Interestingly, mtDNA biogenesis is not sufficient for this proliferation, which also requires the stress-response transcription factor FoxO/DAF-16. At the level of host fitness, FoxO/DAF-16 also prevents food scarcity from accelerating the selection against selfish mtDNA. This suggests that the ability to cope with nutrient stress can promote host tolerance of cheaters. Our study delineates environmental effects on selfish mtDNA dynamics at different levels of selection.

*For correspondence:
maulik.r.patel@vanderbilt.edu

Competing interests: The authors declare that no competing interests exist.

## Introduction

Life is generally organized into a hierarchy of cooperative collectives: multiple genes make up a genome, different genomes combine to form the eukaryotic cell, individual cells give rise to communities and multicellular organisms, and multicellular organisms are often organized into larger groups. New levels of organization emerge when natural selection favors cooperation and the loss of conflict between previously autonomous replicating entities, giving rise to a collective unit upon which selection can further operate (*Michod et al., 2006*; *West et al., 2015*; *Queller and Strassmann, 2009*). Cooperation thus underlies the evolution of larger, more complex biological systems (*Fisher and Regenberg, 2019*; *Gulli et al., 2019*; *Michod et al., 2006*; *West et al., 2015*; *Hammerschmidt et al., 2014*). However, because cooperators incur the near-term cost of contributing to the fitness of others for long-term benefit, cooperation creates opportunities for the emergence of selfish 'cheater' entities, which show up at multiple levels of the hierarchy of biological organization. One type of cheater, meiotic drive genes, facilitate their own transmission by compromising the fitness of gametes lacking them, with examples identified in plants, fungi, and animals (*Bravo Núñez et al., 2018*; *Hammond et al., 2012*; *Hu et al., 2017*; *Larracuente and Presgraves, 2012*; *Schimenti, 2000*). Cancer is characterized by unchecked cell proliferation and the monopolization of resources at the expense of other cells, constituting a form of cheating at the cellular level (*Aktipis et al., 2015*). Cheating behaviors likewise occur among many species of social animals (*Riehl and Frederickson, 2016*).

By benefiting from the contributions of cooperators without reciprocating, cheaters gain a fitness advantage (*Aktipis et al., 2015*; *Dobata et al., 2009*; *Ghoul et al., 2014*; *Strassmann et al., 2000*).

This advantage can break down at higher levels of biological organization, which rely on cooperation at lower levels (*Aktipis et al., 2015*; *de Vargas Roditi et al., 2013*; *Fiegna and Velicer, 2003*; *Moreno-Fenoll et al., 2017*; *Rainey and Rainey, 2003*; *Wenseleers and Ratnieks, 2004*). Hence, selection can simultaneously favor different traits across the levels of the biological hierarchy, a phenomenon known as multilevel selection (*de Vargas Roditi et al., 2013*; *Hammerschmidt et al., 2014*; *Shaffer et al., 2016*; *Takeuchi and Kaneko, 2019*; *Wilson and Wilson, 2007*). Multilevel selection thus provides an explanation for the paradoxical coexistence of selfishness and cooperation in hierarchically structured populations.

Competition over limited resources shapes relative reproductive fitness, and hence Darwinian evolution. Accordingly, efficient resource utilization likely represents an adaptive benefit of cooperative groups (*Koschwanez et al., 2013*; *Vanthournout et al., 2016*). Interestingly, some studies have shown that resource abundance can promote public-goods cooperation, particularly in cases where resource abundance lowers the cost of cooperating (*Brockhurst et al., 2008*; *Connelly et al., 2017*; *Sexton and Schuster, 2017*). Conversely, resource scarcity can promote cooperation (*Cao et al., 2015*; *Chisholm and Firtel, 2004*; *Koschwanez et al., 2013*; *Li and Purugganan, 2011*; *Pereda et al., 2017*; *Requejo and Camacho, 2011*) and abundance can promote selfishness (*Chen and Perc, 2014*; *Ducasse et al., 2015*; *Velicer et al., 1998*). Given that cooperators and cheaters are favored by different levels of selection, taking multilevel selection into account can provide deeper insights into the relationship between resource availability and cooperator-cheater dynamics.

We sought to investigate the relationship between resource availability and multilevel selection using a mitochondrial heteroplasmy. Mitochondria cooperate with each other and with their host by supplying energy; in return, the nuclear genome supplies proteins and building blocks needed to replicate mitochondrial DNA (mtDNA). Mitochondrial organelles can contain multiple copies of mtDNA, which are usually non-recombining and can replicate throughout the cell cycle (*Chatre and Ricchetti, 2013*; *Newlon and Fangman, 1975*; *Sena et al., 1975*). This can give rise to a mixed (heteroplasmic) population of mtDNA variants that compete for transmission. Selfish mtDNA are those that undergo positive selection within hosts and negative selection at the level of host fitness, with examples documented in plants, fungi, and animals (*Havird et al., 2019*; *Klucnika and Ma, 2019*; *Taylor et al., 2002*). Hence, multilevel selection shapes the population dynamics of mitochondria (*Dubie et al., 2020*; *Havird et al., 2019*; *Klucnika and Ma, 2019*; *Shou, 2015*; *Taylor et al., 2002*).

How does resource availability shape the multilevel selection forces acting on selfish mtDNA? We address this using the model species *Caenorhabditis elegans* harboring the well-characterized heteroplasmic mutant genome *uaDf5* (*Figure 1A,B*), hereinafter referred to as ∆mtDNA (*Ahier et al., 2018*; *Gitschlag et al., 2016*; *Liau et al., 2007*; *Lin et al., 2016*; *Tsang and Lemire, 2002*). This deletion mutation spans four protein-coding genes and seven tRNA genes (*Figure 1B*), disrupting gene expression and metabolic function. Previous work has shown that ∆mtDNA propagates at the expense of host fitness by exploiting regulatory mechanisms encoded by the host genome. For example, the copy number of the mutant genome increases in addition to—rather than at the expense of—wildtype mtDNA copy number (*Gitschlag et al., 2016*), suggesting that ∆mtDNA hitchhikes to higher levels by evading the host's ability to regulate mtDNA copy number. The presence of ∆mtDNA also elicits the activation of host stress-response genes, inadvertently promoting further ∆mtDNA proliferation (*Gitschlag et al., 2016*; *Lin et al., 2016*). Here, we sought to expand the investigation of this biological cheater to include the ecologically relevant context of resource availability.

First, we isolate and measure selection on ∆mtDNA separately within individual hosts (sub-organismal) and between hosts (organismal). We then adapt the multilevel selection framework to study the effects of food availability and the physiology of nutrient stress tolerance on ∆mtDNA. Although diet and nutrient sensing govern overall mtDNA levels, the preferential proliferation of the selfish genome at the sub-organismal level depends on a key regulator of stress tolerance, namely the Forkhead box O (FoxO) transcription factor DAF-16. Diet restriction strengthens organismal selection against the selfish mtDNA, but only in the absence of DAF-16. We conclude that food availability and resilience to food scarcity govern the relative fitness of the cooperators and cheaters both within and between collectives.

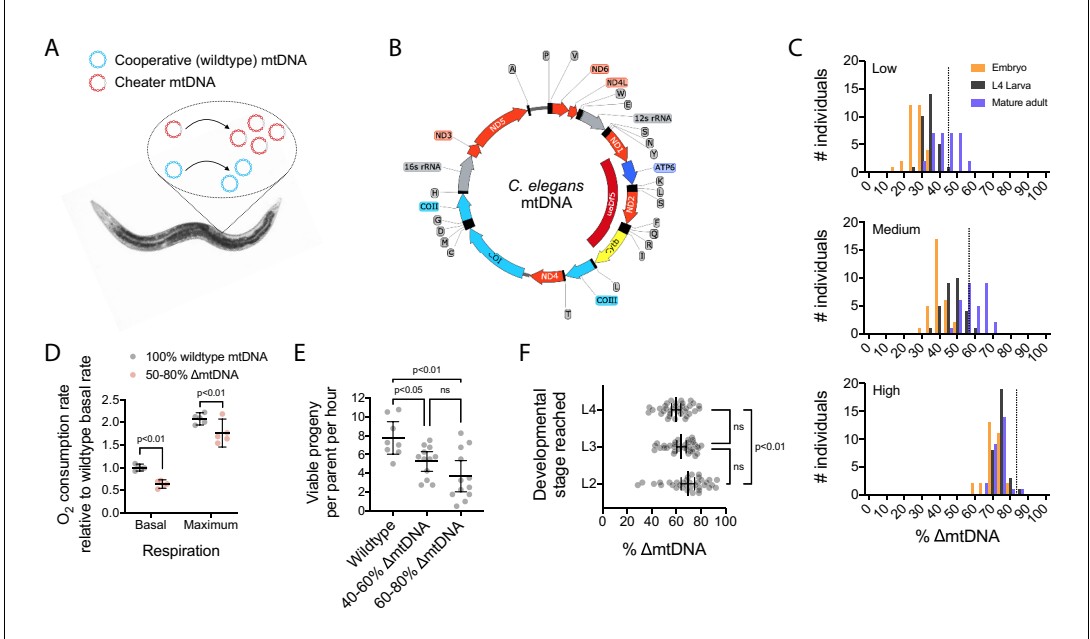

**Figure 1.** The *uaDf5* mutant variant (ΔmtDNA) proliferates despite undermining host fitness, indicative of a cheater undergoing multilevel selection. (**A**) Selfish mtDNA behaves as a biological cheater, outcompeting the cooperative wildtype mtDNA within hosts. (**B**) *C. elegans* mtDNA map showing *uaDf5* deletion (dark red) in ΔmtDNA and color-coded genes: respiratory complex I (light red), complex III (yellow), complex IV (light blue), complex V (dark blue), ribosomal RNA (gray), tRNA (black), non-coding regions (thin line). (**C**) ΔmtDNA frequency across developmental stages of single broods from low (top, N = 94), intermediate (middle, N = 93), or high (bottom, N = 88) parental ΔmtDNA frequency (dotted lines). Mature adults were lysed at day 2 of adulthood, the same age at which the parents were lysed. (**D**) Basal and maximum aerobic respiration in age-synchronized L4 animals. Two-way ANOVA with Sidak's multiple comparisons test. (**E**) Peak fecundity (viable progeny per hour per parent at day 2 of adulthood) binned according to the low end of the ΔmtDNA frequency distribution (below the population mean of 60%, N = 12) or the high end (above 60%, N = 12), with wildtype controls (N = 8). Brown-Forsythe and Welch ANOVA with Dunnett's T3 multiple comparisons test. (**F**) Larval stage reached within 48 hr starting from age-synchronized embryos, plotted as a function of ΔmtDNA frequency. N = 35 nematodes per larval stage. Brown-Forsythe and Welch ANOVA with Dunnett's T3 multiple comparisons test. All experiments featured in this figure used nematodes that were maintained on a diet of live OP50 *E. coli* at 20 °C.

The online version of this article includes the following figure supplement(s) for figure 1:

**Figure supplement 1.** Quantification of mtDNA copy number and ΔmtDNA frequency by droplet digital PCR.

# Results

## An experimental strategy to isolate selection on a selfish mitochondrial genome at different levels

Selection can act directly on individual mtDNA molecules within an organelle due to intrinsic replication advantage (*Holt et al., 2014*). Selection can also occur between organelles within a host cell (*Lieber et al., 2019*; *Zhang et al., 2019*), between cells within a multicellular host (*Shidara et al., 2005*), and finally between host organisms. By focusing on selection for mitochondrial genotype itself, we bypass the challenges facing the study of selection acting on organelles, which undergo fusion and fission dynamics and hence do not exist as discrete units. Moreover, the vast majority of mitochondrial content in the adult hermaphroditic nematode *Caenorhabditis elegans* is confined to the germline (*Bratic et al., 2009*), which exists as a contiguous syncytium of cytoplasm until the final stages of oocyte maturation (*Pazdernik and Schedl, 2013*). Sub-organismal selection thus predominantly reflects the biology of the female germline, where mtDNA variants compete for transmission (*Figure 1A*). Accordingly, here we focus on selection for mitochondrial genotype at the sub-organismal level as a single phenomenon, in addition to selection for mitochondrial genotype at the organismal level.

Using a multiplex droplet digital PCR (ddPCR) approach to quantify mitochondrial genotype (*Figure 1—figure supplement 1A–C*), we observed that ΔmtDNA frequency steadily rises across

organismal development in a manner that depends on the initial inherited frequency of ΔmtDNA (*Figure 1C*), consistent with earlier work (*Tsang and Lemire, 2002*). The apparent upper limit of sub-organismal ΔmtDNA proliferation is indicative of the phenomenon of frequency-dependent selection, a common feature of cheater entities (*Dobata and Tsuji, 2013*; *Dugatkin et al., 2005*; *Pruitt and Riechert, 2009*; *Riehl and Frederickson, 2016*; *Ross-Gillespie et al., 2007*). Another important feature of cheaters is that their selection advantage tends to break down at higher levels of selection (*Aktipis et al., 2015*; *de Vargas Roditi et al., 2013*; *Fiegna and Velicer, 2003*; *Moreno-Fenoll et al., 2017*; *Rainey and Rainey, 2003*; *Wenseleers and Ratnieks, 2004*). Interestingly, although host stress-response mechanisms have previously been implicated in ΔmtDNA propagation (*Gitschlag et al., 2016*; *Lin et al., 2016*), such mechanisms do not appear to protect the host from the fitness cost incurred by harboring ΔmtDNA. On the contrary, we observed several indicators that ΔmtDNA proliferates while compromising host fitness, consistent with multilevel selection and in agreement with prior studies of this genome (*Gitschlag et al., 2016*; *Liau et al., 2007*; *Lin et al., 2016*). Indicators of reduced host fitness include reduced aerobic respiration (*Figure 1D*), in spite of elevated overall mitochondrial mass and the activation of mitochondrial stress-response mechanisms (*Gitschlag et al., 2016*; *Lin et al., 2016*). Other indicators that ΔmtDNA impacts host fitness include reduced fertility (*Figure 1E*) and slowed development (*Figure 1F*) in heteroplasmic animals. We therefore sought to quantitatively characterize the multilevel selection dynamics of ΔmtDNA.

To measure the impact of sub-organismal selection on the propagation of ΔmtDNA across generations, ΔmtDNA frequency was quantified longitudinally at successive developmental stages and across multiple parent-progeny lineages. Individual parent-progeny lineages were maintained in isolation from one another to minimize the confounding effect of organismal selection on ΔmtDNA frequency. Initially, we observed reduced ΔmtDNA frequency in embryos compared to their parents (*Figure 2A*), consistent with the notion of germline purifying selection (*Ahier et al., 2018*; *Hill et al., 2014*; *Lieber et al., 2019*; *Ma et al., 2014*; *Stewart et al., 2008*). However, ΔmtDNA proliferates across development, achieving even higher frequency on average in adult progeny than in their respective parents (*Figure 2A*). Moreover, the magnitude of this proliferation declines with increasing parental ΔmtDNA frequency (*Figure 2B*), consistent with the phenomenon of negative frequency-dependent selection. Overall, we have isolated and quantitatively measured the impact of selection at the sub-organismal level on ΔmtDNA propagation across generations.

To measure selection against ΔmtDNA strictly at the level of host fitness, we competed heteroplasmic animals carrying ΔmtDNA against their homoplasmic wildtype counterparts on the same food plate (*Figure 2C*). In parallel, we propagated non-competing control lines, which lacked wildtype animals. Consistent with organismal selection, we observed a decline in the fraction of individuals carrying ΔmtDNA across all eight replicate lineages (*Figure 2D* and *Figure 2—figure supplement 1A*). We also quantified ΔmtDNA frequency directly across all competing and non-competing lines using multiplex ddPCR, which revealed a dramatic decline in ΔmtDNA frequency across all eight competing lines (*Figure 2—figure supplement 1B*). This decline was not observed in the non-competing lines, which maintained ΔmtDNA frequency near 60% despite minor variation (*Figure 2—figure supplement 1B*). In order to isolate the effect of organismal selection on ΔmtDNA, we controlled for the confounding factors such as sub-organismal ΔmtDNA dynamics. To accomplish this, the ΔmtDNA frequencies of the competing lines were normalized to that of the non-competing lines at each generation. This effectively normalizes population frequency to individual (sub-organismal) frequency, since non-competing lines contain only ΔmtDNA-carrying individuals and thus their frequency is equal to average sub-organismal frequency. Moreover, normalizing to the non-competing lines sets the slope of ΔmtDNA frequency of those lines to zero (*Figure 2E*, gray lines). Then, whatever non-zero slope remains for the competing lines (*Figure 2E*) can be attributed to the presence of homoplasmic wildtype individuals (the only variable distinguishing the competing from non-competing lines). In conclusion, we have separately measured the effects of selection on ΔmtDNA at the sub-organismal and organismal levels, which we propose balance to allow the stable persistence of ΔmtDNA.

## Nutrient availability influences sub-organismal ΔmtDNA dynamics

We next sought to investigate how resource availability affects the multilevel selection dynamics of ΔmtDNA. Nematodes were raised on food plates seeded with either a high or low concentration of *E. coli* (OP50 strain), which were UV-killed to prevent further bacterial growth (*Figure 3—figure*

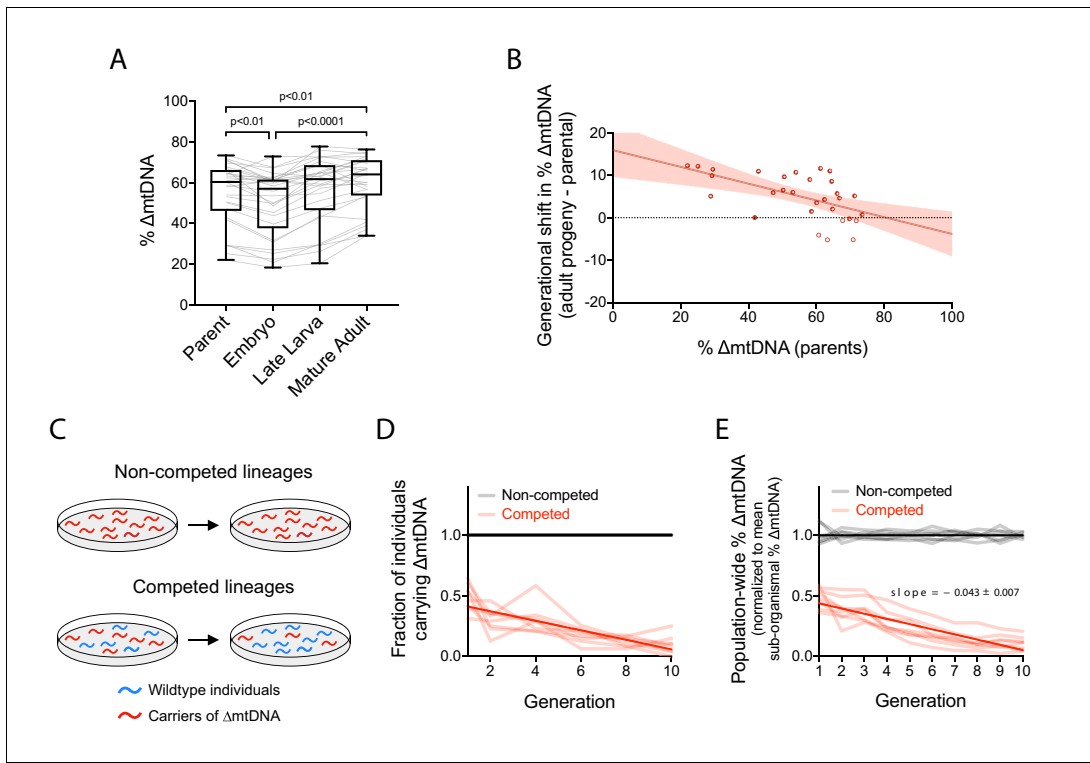

**Figure 2.** Quantification of intergenerational changes in ΔmtDNA frequency due to selection at sub-organismal and organismal levels. (**A**) ΔmtDNA frequency across parent-progeny lineages, maintained in isolation to minimize the effect of organismal selection. Each light gray line represents a single lineage consisting of a parent lysed individually followed by 3 of its progeny pooled and lysed together at each of 3 developmental time-points. Mature adults were lysed at day 2 of adulthood, the same age at which the parents were lysed, to ensure that parents and their adult progeny were age-matched. Box and whisker plots depict mean ΔmtDNA frequency and each quartile. Friedman test with Dunn's multiple comparisons test. N = 30 lineages. (**B**) Shift in ΔmtDNA frequency per generation, obtained by subtracting ΔmtDNA frequency of mature adult progeny in panel (**A**) from ΔmtDNA frequency of the respective parent, plotted as a function of parental ΔmtDNA frequency. Red shaded region: 95% C.I. (**C**) Competition experiment designed to quantify organismal selection against ΔmtDNA. (**D**) Fraction of ΔmtDNA-carrying heteroplasmic individuals across 10 generations and eight replicate competed (red) versus non-competed (gray) lineages. Competed lineages consisted of heteroplasmic and homoplasmic wildtype individuals on the same food plates. Non-competed lineages consisted of heteroplasmic individuals only. Solid lines represent best-fit regressions across all replicate lineages. (**E**) Population-wide ΔmtDNA frequency across the organismal competition experiment. To isolate the change in ΔmtDNA frequency that occurs strictly due to organismal selection, we controlled for the confounding influence of sub-organismal ΔmtDNA dynamics by normalizing the total ΔmtDNA frequency of each lineage to the mean frequency of the non-competed lines at each generation. Because the non-competed lines consist entirely of heteroplasmic individuals, ΔmtDNA frequency across the non-competed lines is equal to mean sub-organismal ΔmtDNA frequency. The overall slope of the non-competing lines is therefore set to zero and the non-zero slope across the competing lines is due to the presence of wildtype animals (see panel C), allowing us to measure the effect of organismal selection by itself. Solid lines represent best-fit regressions across all replicate lineages. All experiments featured in this figure used nematodes that were maintained on a diet of live OP50 *E. coli* at 20 °C. Error bars: 95% C.I.

The online version of this article includes the following source data and figure supplement(s) for figure 2:

**Source data 1.** Summary statistics for frequency-dependent change in ΔmtDNA frequency at sub-organismal level and organismal selection against ΔmtDNA.

**Figure supplement 1.** Change in ΔmtDNA frequency in competition experiments is due to organismal selection.

*supplement 1A,B*). Although UV-killed OP50 partially mimics diet restriction (*Win et al., 2013*), we found that nematodes raised on the more restricted (low concentration) diet harbored significantly lower ΔmtDNA frequency compared to those raised on a more abundant (high concentration) control diet (*Figure 3A*). Moreover, sub-organismal ΔmtDNA frequency is even higher in animals raised

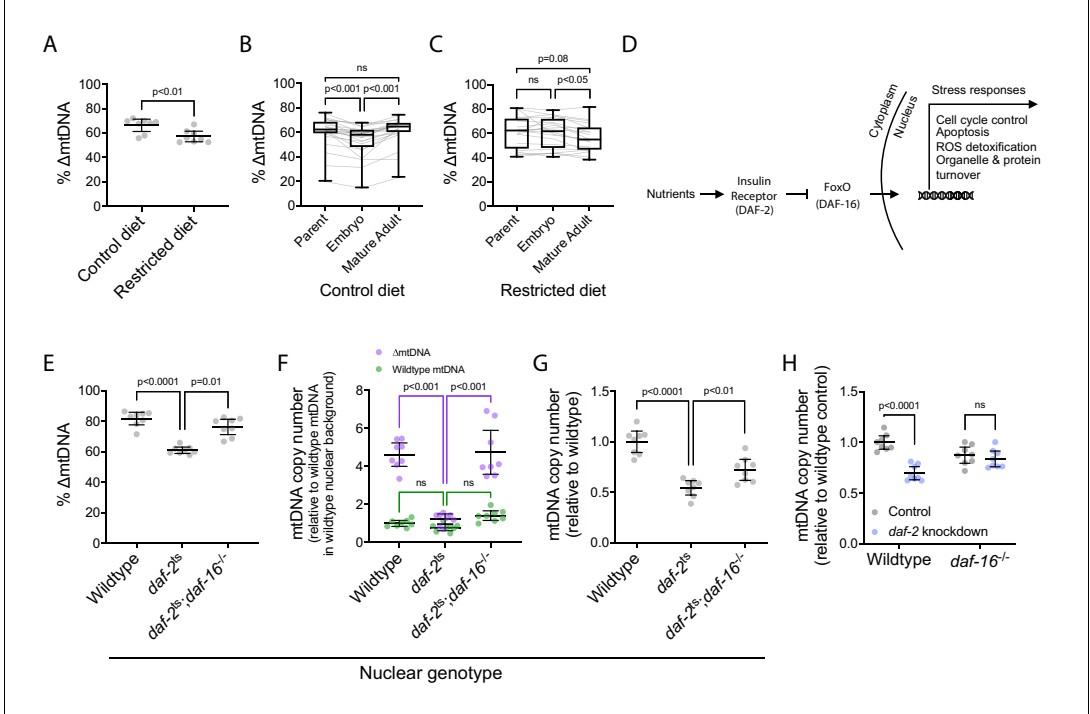

**Figure 3.** ΔmtDNA exploits nutrient supply and insulin signaling to proliferate at the sub-organismal level. (**A**) ΔmtDNA frequency on restricted versus control diet. Nematodes were maintained at 20 °C on a diet of UV-killed OP50 *E. coli*. Mann-Whitney test. N = 8 pooled lysates of 5 age-synchronized day-4 adults each. (**B–C**) ΔmtDNA frequency across parent-progeny lineages raised on control (**B**) or restricted (**C**) diet. Each light gray line represents a single lineage consisting of a parent lysed individually followed by 3 of its progeny pooled and lysed together at each of 2 developmental time-points. Nematodes were maintained at 20 °C on a diet of UV-killed OP50 *E. coli*. Mature adults were lysed at day 2 of adulthood, the same age at which the parents were lysed, to ensure that parents and their adult progeny were age-matched. Box and whisker plots depict mean ΔmtDNA frequency and each quartile. Friedman test with Dunn's multiple comparisons test. N = 24 lineages (control diet); N = 20 lineages (restricted diet). (**D**) FoxO-dependent insulin signaling cascade with *C. elegans* homologs in parentheses. (**E**) ΔmtDNA frequency among wildtype, *daf-2*(e1370) mutant, and *daf-2* (e1370);*daf-16*(mu86) double-mutant host genotypes, on a plentiful diet consisting of live OP50 *E. coli*. Kruskal-Wallis ANOVA with Dunn's multiple comparisons test. N = 8 pooled lysates of 5 age-synchronized day-4 adults each. (**F**) mtDNA copy number of individuals in (**E**), normalized to wildtype mtDNA from wildtype nuclear background. Green and purple represent wildtype and ΔmtDNA copy number, respectively. Two-way ANOVA with Sidak's multiple comparisons test. N = 8 pooled lysates of 5 age-synchronized day-4 adults each. (**G**) mtDNA copy number in homoplasmic adults of wildtype, *daf-2*(e1370) mutant, and *daf-2*(e1370);*daf-16*(mu86) double-mutant host genotypes, on a plentiful diet consisting of live OP50 *E. coli*. One-way ANOVA with Dunnett's multiple comparisons test. N = 8 pooled lysates of 5 age-synchronized day-4 adults each. (**H**) mtDNA copy number in homoplasmic adults lacking ΔmtDNA of either wildtype or null *daf-16*(mu86) host genotype, on either *daf-2* RNAi knockdown or empty-vector control conditions. Two-way ANOVA with Sidak's multiple comparisons test. N = 8 lysates containing five pooled age-synchronized day-4 adults each. Experiments depicted in panels (**E**) through (**H**) used nematodes that were maintained at 16 °C during larval development and transferred at the L4 stage to 25 °C for adult maturation, corresponding to the permissive and restrictive temperatures for the *daf-2*(e1370) allele, respectively. Error bars: 95% C.I.

The online version of this article includes the following figure supplement(s) for figure 3:

**Figure supplement 1.** Diet and insulin signaling regulate mtDNA copy number and ΔmtDNA frequency.

on live food (*Figure 3—figure supplement 1C*). Consistent with this observation, we notice a greater net shift in ΔmtDNA frequency from parent to adult progeny on live food (*Figure 2A*) compared to UV-killed food (*Figure 3B*). Despite the attenuated ΔmtDNA proliferation on UV-killed food, we still observe noteworthy dietary effects that result simply by varying the concentration of the UV-killed food. First, the initial selection against ΔmtDNA between parent and embryo was abolished by diet restriction, corresponding to a 100-fold dilution of the control diet (*Figure 3B,C*). Second, ΔmtDNA frequency rose from embryos to adults on a control diet, recovering from the initial purifying selection between parent and embryo (*Figure 3B*), but failed to do so in animals grown on a restricted diet (*Figure 3C*). These observations reveal complex, life-stage-specific effects of diet on

ΔmtDNA dynamics: a plentiful diet selects against ΔmtDNA between parent and embryo but selects for ΔmtDNA across development.

## ΔmtDNA exploits nutrient sensing to proliferate across development

To better understand the role of nutrient status in the cheating behavior of ΔmtDNA, we focused on sub-organismal ΔmtDNA proliferation across development. In particular, we hypothesized that the insulin-signaling pathway underlies ΔmtDNA proliferation. Insulin acts as a nutrient-dependent growth hormone and regulator of metabolic homeostasis (*Figure 3D*), tailoring the appropriate physiological responses to external nutrient conditions (*Badisco et al., 2013*; *Danielsen et al., 2013*; *Lee and Dong, 2017*; *Lopez et al., 2013*; *Michaelson et al., 2010*; *Puig and Tjian, 2006*; *Shiojima et al., 2002*; *Das and Arur, 2017*; *Porte et al., 2005*). Nematodes expressing a defective allele of the insulin receptor homolog *daf-2* perceive starvation, even in presence of food. However, disrupting insulin signaling in young larvae causes the dauer phenotype, a form of developmental arrest (*Gottlieb and Ruvkun, 1994*). Thus, we used a temperature-sensitive *daf-2* allele to conditionally inactivate insulin signaling. Animals were incubated at a permissive temperature (16 °C) to preserve insulin signaling during early larval development, thereby preventing developmental arrest. Animals were transferred to the restrictive temperature (25 °C) beginning at the last larval stage (L4) to inactivate insulin signaling during adult maturation when most mtDNA replication occurs. Compared to animals with intact insulin signaling, we observed lower ΔmtDNA frequency in animals expressing the defective *daf-2* allele (*Figure 3E*). This difference was absent in control lines that were maintained at the permissive temperature of 16 °C (*Figure 3—figure supplement 1D*), suggesting that loss of insulin signaling limits ΔmtDNA proliferation. Moreover, no overall change in ΔmtDNA frequency occurred across four independent lineages of *daf-2* mutants even after four consecutive generations (*Figure 3—figure supplement 1E*). In contrast, ΔmtDNA frequency increased substantially in wildtype controls. These data show that nutrient sensing via the insulin-signaling pathway is involved in sub-organismal proliferation of ΔmtDNA.

The insulin receptor communicates nutrient status to the cell largely through the negative regulation of the FoxO family of transcription factors (*O-Sullivan et al., 2015*), encoded by the gene *daf-16* in *C. elegans* (*Figure 3D*). Nutrient limitation or inactivation of the receptor activates FoxO/DAF-16, resulting in altered expression of its target genes. Interestingly, deletion of *daf-16* restores the proliferation of ΔmtDNA in animals defective for *daf-2* function (*Figure 3E*). We conclude that insulin signaling promotes sub-organismal ΔmtDNA proliferation through the protein DAF-16.

How does DAF-16-dependent insulin signaling affect ΔmtDNA proliferation? The reduction of ΔmtDNA frequency by DAF-2 inactivation, and the rescue of ΔmtDNA frequency by loss of DAF-16 (*Figure 3E*), are almost entirely attributable to large differences in the copy number of ΔmtDNA but not of wildtype mtDNA (*Figure 3F*). In other words, insulin signaling promotes elevated total mtDNA copy number, perhaps as a driver of ΔmtDNA proliferation or as a consequence of it. To distinguish between these possibilities, we quantified copy number in animals lacking ΔmtDNA. Homoplasmic wildtype mtDNA copy number was quantified using the multiplex ddPCR method. To obtain relative copy number, raw mtDNA copy number across each nuclear genotype was normalized to that of the wildtype controls. In homoplasmic animals, we observed lower mtDNA copy number upon loss of insulin signaling, whether by *daf-2* mutation (*Figure 3G* and *Figure 3—figure supplement 1G*) or by knockdown of *daf-2* gene expression (*Figure 3H*), consistent with previous work in *Drosophila* (*Wang et al., 2019*). Loss of DAF-16 partially but significantly rescued copy number (*Figure 3G,H* and *Figure 3—figure supplement 1G*). Together, these data suggest that DAF-2 signaling inhibits DAF-16 to allow high mtDNA copy number, which permits sub-organismal ΔmtDNA proliferation.

How does DAF-16 suppress mtDNA copy number? This suppression could be achieved via mechanisms that result in the elimination of mitochondria, either at the organelle level through increased mitochondrial autophagy (mitophagy) or at the cellular level through increased apoptosis. Consistent with these possibilities, previous studies have identified FoxO/DAF-16 as a regulator of genes involved in autophagy and apoptosis (*Murtaza et al., 2017*; *Webb and Brunet, 2014*; *Webb et al., 2016*). We therefore reasoned that upon loss of insulin signaling, DAF-16 might suppress mtDNA copy number by upregulating either the destruction of mitochondrial organelles or cell death in the female germline. To test this idea, we genetically targeted PINK1/Parkin-dependent mitophagy using a deletion of *pdr-1*, encoding the *C. elegans* Parkin homolog. We genetically targeted

apoptosis with a deletion of *ced-3*, encoding the terminator caspase in *C. elegans*. Disrupting either of these processes did not restore mtDNA copy number in *daf-2* mutants (*Figure 4A,B*), nor did we observe increased germline apoptosis in *daf-2* mutants (*Figure 4—figure supplement 1A,B*). Because mitochondrial degradation can occur in a PINK1/Parkin-independent manner (*Allen et al., 2013*; *Di Rita et al., 2018*; *Hibshman et al., 2018*), we therefore also tested for a potential role of mitochondrial fission, a common precursor of mitophagy, using a deletion in the gene *drp-1*, which encodes a dynamin-related protein important for mitochondrial fission. Although we observed an increase in mtDNA copy number when mitochondrial fission is disrupted in animals with intact insulin signaling (*Figure 4C*), consistent with reduced mitophagy, we did not observe a rescue of mtDNA copy number in *daf-2*These data suggest that the suppression of mtDNA content upon loss of insulin signaling is not mediated through the elimination of mitochondria by PINK/Parkin-dependent mitophagy, mitochondrial fission, or apoptosis.

Alternatively, DAF-16 might restrict mtDNA biogenesis. Nutrient availability and insulin signaling each promote development of the female germline (*Angelo and Van Gilst, 2009*; *Drummond-Barbosa and Spradling, 2001*; *Michaelson et al., 2010*; *Narbonne and Roy, 2006*; *Shim et al., 2002*), which harbors the vast majority of mtDNA in the adult nematode (*Bratic et al., 2009*). We observed that mitochondrial organelle quantity and mtDNA copy number are proportional to gonad size and cell count, respectively, across wildtype, *daf-2* mutant, and *daf-2;daf-16* double-mutants (*Figure 4D–G*). We therefore conclude that suppression of germline development by DAF-16 accounts for the reduced mtDNA content in insulin-signaling mutants (*Figure 4H*).

Because DAF-16 is required for mtDNA copy-number suppression upon loss of insulin signaling, we reasoned that DAF-16 should also be required for copy-number suppression in response to diet restriction. However, while diet restriction suppresses mtDNA copy number, this occurs independently of DAF-16 (*Figure 5A*). Given that ΔmtDNA frequency is sensitive to changes in total mtDNA copy number (*Figure 3E–G*), the effect of diet on total copy number suggests that diet might also modulate ΔmtDNA frequency independently of DAF-16. Remarkably, we only saw diet-dependent elevation in ΔmtDNA frequency when DAF-16 was present (*Figure 5B*). Moreover, while total mtDNA copy number and ΔmtDNA frequency each rose significantly across development on a control relative to restricted diet (*Figure 5C*), copy number rose by itself, with no accompanying change in ΔmtDNA frequency, in *daf-16* mutants (*Figure 5D*). Because diet restriction and loss of DAF-16 were each found to attenuate ΔmtDNA proliferation, we conclude that nutrient abundance and DAF-16 are each necessary, but not sufficient individually, for ΔmtDNA to maintain a sub-organismal selection advantage.

## Nutrient status governs selection on ΔmtDNA at different levels

FoxO/DAF-16 regulates numerous genes involved in stress tolerance (*Klotz et al., 2015*; *Martins et al., 2016*; *Murphy et al., 2003*; *Tepper et al., 2013*; *Webb et al., 2016*) and promotes organismal survival during nutrient scarcity (*Greer et al., 2007*; *Hibshman et al., 2017*; *Kramer et al., 2008*). We therefore asked whether nutrient availability and DAF-16 affect selection on ΔmtDNA at both the organismal and sub-organismal levels. Sub-organismal selection was quantified as before (see *Figure 2B*), under restricted versus control diets, in the presence versus absence of DAF-16 (*Figure 6A,B*). Organismal selection was quantified under each of these same conditions, using the competition method previously described (see *Figure 2C–E*). In populations with wildtype DAF-16, diet restriction did not significantly affect the decline in ΔmtDNA frequency at the level of organismal selection (*Figure 6C,D* and *Figure 6—figure supplement 1A*). However, diet restriction accelerated the decline of ΔmtDNA frequency at the level of organismal selection among *daf-16* mutants (*Figure 6E,F* and *Figure 6—figure supplement 1B*). These data indicate that although food scarcity can strengthen selection against ΔmtDNA at the organismal level, DAF-16 protects ΔmtDNA from this effect.

Finally, we sought to integrate our observations of sub-organismal and organismal selection for each of the four conditions tested. We observed that the sub-organismal selection advantage of ΔmtDNA is compromised by diet restriction, loss of DAF-16, or both (*Figures 5*, *6A, B*). Furthermore, diet restriction was observed to accelerate organismal selection against ΔmtDNA, but only in the absence of DAF-16 (*Figure 6C–F* and *Figure 6—figure supplement 1*). Taken together, these observations predict that the strongest net selection against ΔmtDNA occurs in populations lacking DAF-16 and experiencing food scarcity (*Figure 6G*), and the weakest overall

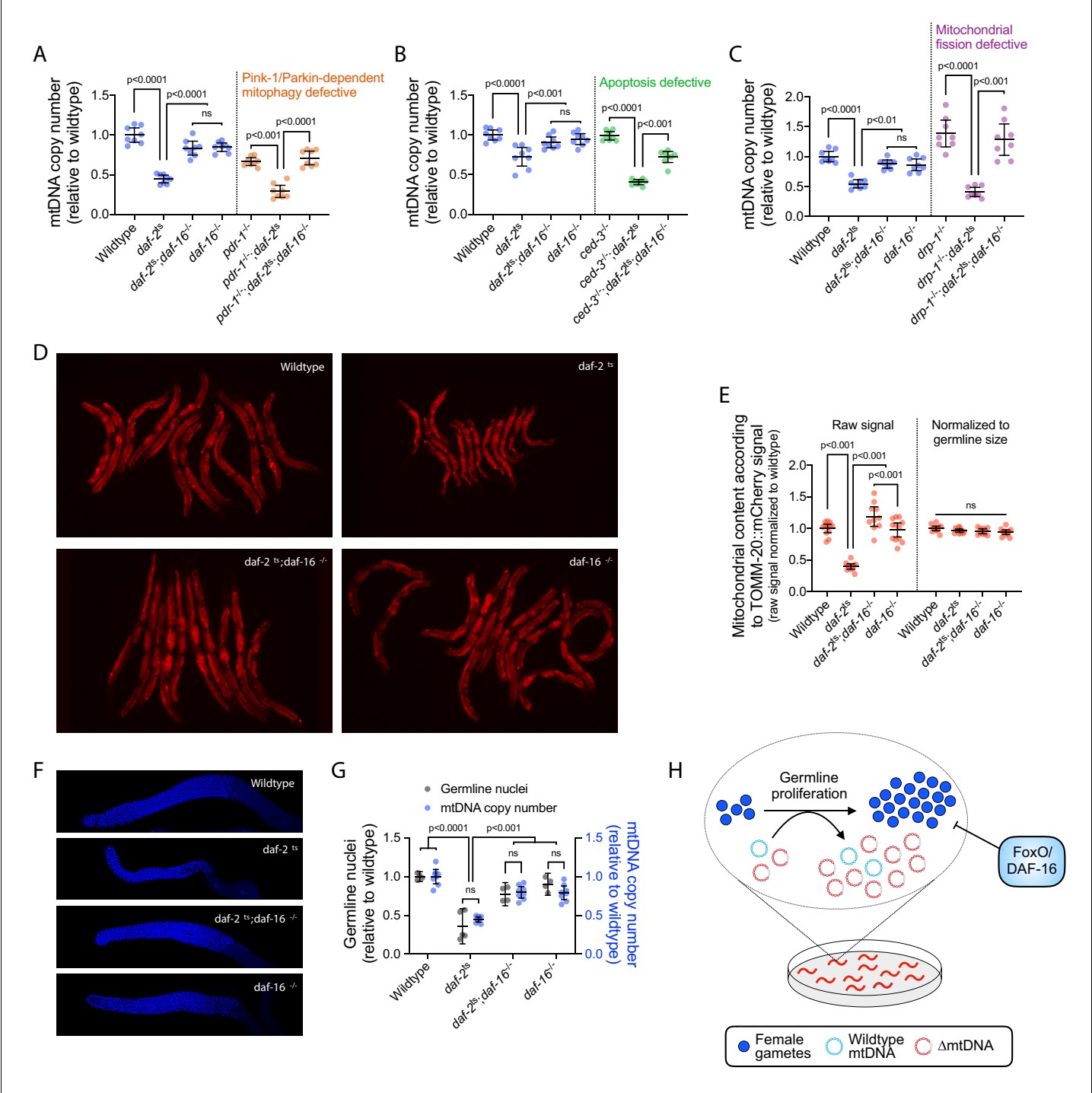

**Figure 4.** DAF-16 activation upon loss of insulin signaling suppresses mtDNA content via regulation of germline proliferation. (A–C) mtDNA copy number in age-synchronized adults of wildtype, temperature-sensitive *daf-2*(e1370) mutant, null *daf-16*(mu86) mutant, or double-mutant genotype. Copy number is also shown in wildtype, *daf-2*(e1370), and *daf-2*(e1370);*daf-16*(mu86) double-mutant adults each paired with *pdr-1*(gk448) (A), *ced-3* (ok2734) (B), or *drp-1*(tm1108) (C), representing loss-of-function alleles of the Parkin homologue, the terminator caspase CED-3, or dynamin-related protein, respectively. Copy number in *daf-16*(mu86) single-mutants is also shown. One-way ANOVA with Sidak's multiple comparisons test. N = 8 lysates containing five pooled age-synchronized day-4 adults each. (D–E) Images (D) and quantification (E) of germline mitochondria labeled with TOMM-20::mCherry across wildtype, *daf-2*(e1370), *daf-16*(mu86), or double-mutant genotype. Each data point in (E) represents one adult in (D). One-way ANOVA with Sidak's multiple comparisons test. (F–G) Representative images (F) and quantification (G) of DAPI-stained nuclei with mtDNA copy number across wildtype, *daf-2*(e1370), *daf-16*(mu86), or double-mutant genotype. Each gray data point represents one adult female gonad. For mtDNA

*Figure 4 continued on next page*

*Figure 4 continued*

copy number, N = 8 pooled lysates of 5 age-synchronized adults each. Two-way ANOVA with Sidak's multiple comparisons test. (**H**) Schematic showing that upon loss of insulin signaling, FoxO/DAF-16 limits ΔmtDNA proliferation by restricting germline development. All experiments featured in this figure used nematodes that were maintained on a diet of live OP50 *E. coli* at 16 °C during larval development and transferred at the L4 stage to 25 °C for adult maturation, corresponding to the permissive and restrictive temperatures for the *daf-2*(e1370) allele, respectively. For panels (**D**) through (**G**), imaging was conducted on day-2 adults to visualize germlines at peak fecundity (*Hughes et al., 2007*). Error bars: 95% C.I.

The online version of this article includes the following figure supplement(s) for figure 4:

**Figure supplement 1.** Germline apoptosis in wildtype and *daf-2* mutant genotypes.

selection occurs in populations with DAF-16 and experiencing food abundance, with the remaining two conditions each experiencing an intermediate strength of selection. Measuring ΔmtDNA frequency across non-competing heteroplasmic populations afforded the opportunity to test this prediction. Remarkably, this prediction is consistent with our observation (*Figure 6H*), even though UV-killed food compromises ΔmtDNA propagation even in the control diet (*Figure 6H*, compare gray to dotted brown line). Combined, our data reveal numerous ways in which diet and host genotype interact to shape the multilevel selection dynamics of a cheater genome (*Figure 7*).

## Discussion

Multilevel selection offers a powerful explanatory framework to understand cooperator-cheater dynamics. However, investigations of multilevel selection face the challenge of trying to account for the confounding influence of selection acting at one level while estimating the strength of selection at a different level (*Goodnight, 2015*; *Goodnight et al., 1992*; *Heisler and Damuth, 1987*). To

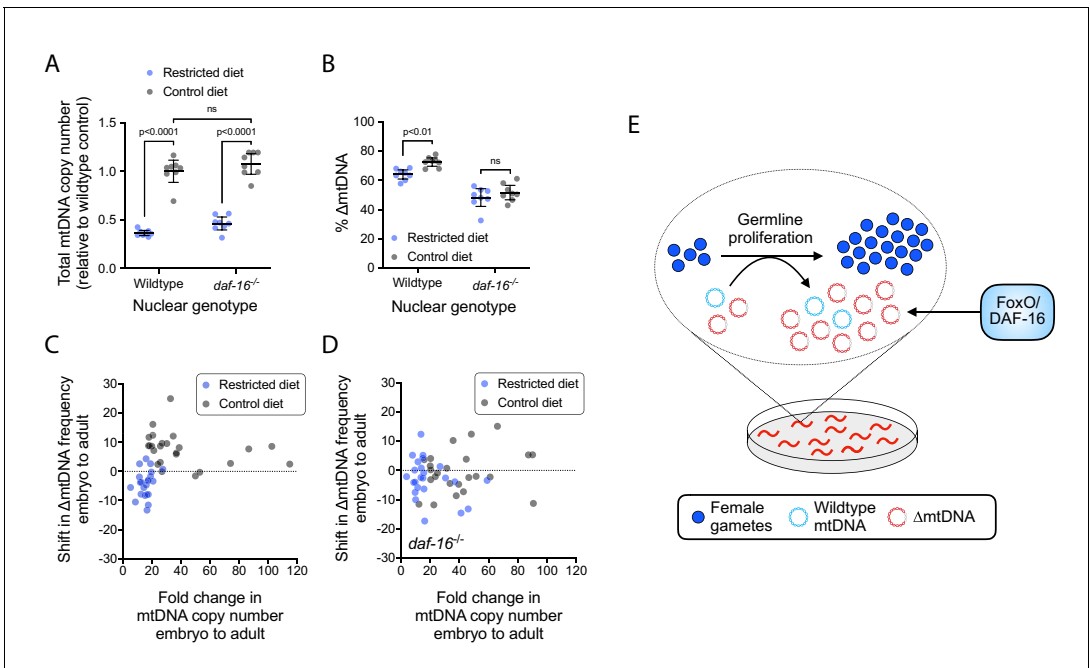

**Figure 5.** The sub-organismal selection advantage of ΔmtDNA requires both nutrient abundance and DAF-16. (**A**) Total mtDNA copy number in heteroplasmic individuals, wildtype versus null *daf-16*(mu86) host genotype, restricted versus control diet. N = 8 pooled lysates of 5 age-synchronized day-4 adults each. Two-way ANOVA with Bonferroni correction. (**B**) ΔmtDNA frequency of individuals in (**A**). Two-way ANOVA with Bonferroni correction. (**C–D**) Change in mtDNA copy number and ΔmtDNA frequency across development, with wildtype (**C**) versus null *daf-16*(mu86) (**D**) host genotype, on restricted versus control diet. Each data point represents the difference in copy number (horizontal axis) and ΔmtDNA frequency (vertical axis) between three pooled day-2 adults (age-matched to their respective parents) and three pooled embryos of the same brood. Mann-Whitney tests with Bonferroni correction. N = 22 wildtype, restricted diet; N = 24 wildtype, control diet; N = 24 *daf-16*(mu86), restricted diet; N = 24 *daf-16*(mu86), control diet. (**E**) Schematic showing that FoxO/DAF-16 is required in order for ΔmtDNA to take advantage of the increased mtDNA replication on an abundant diet. All experiments featured in this figure used nematodes maintained on a diet of UV-killed OP50 *E. coli* at 20 °C. Error bars: 95% C.I.

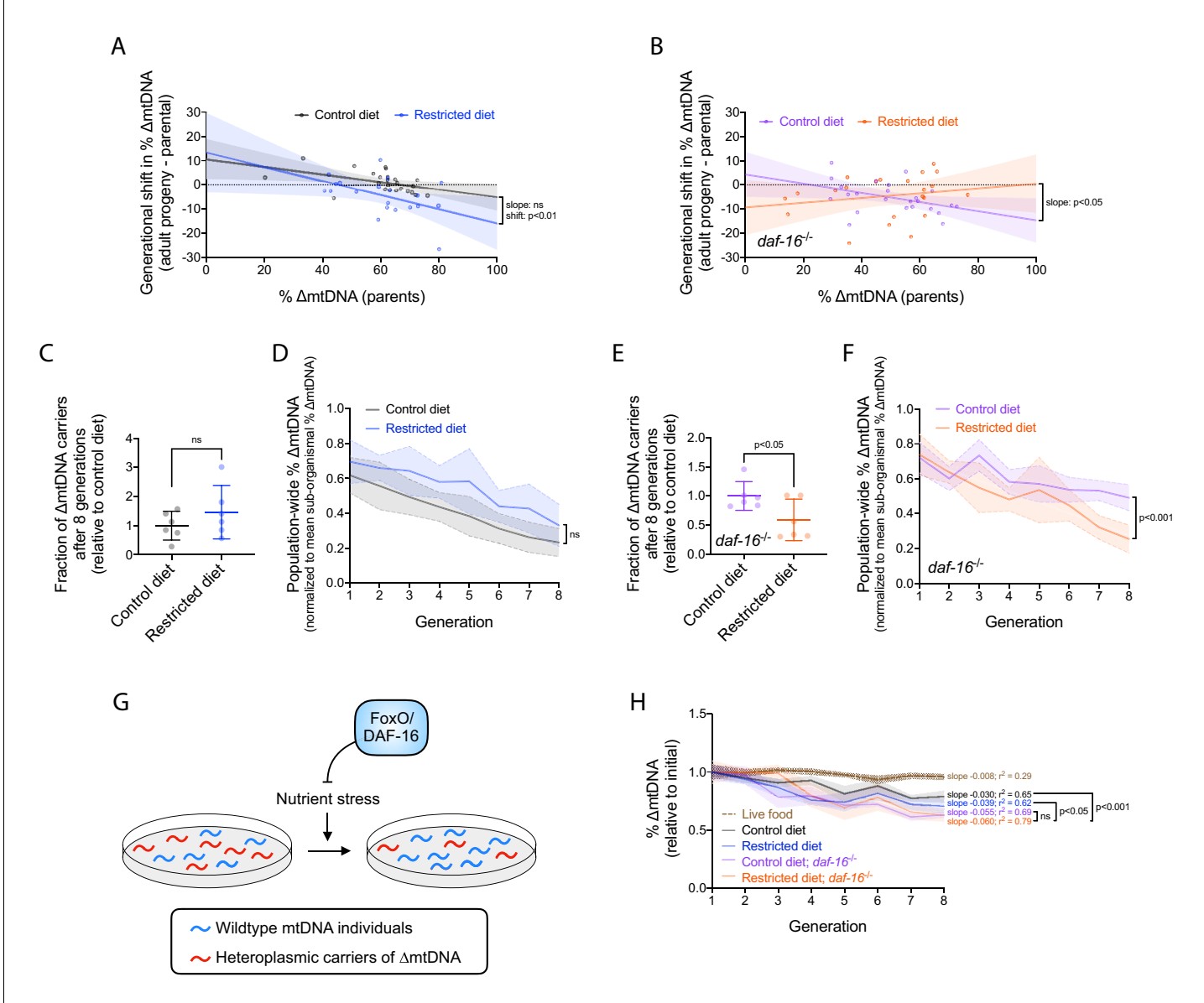

**Figure 6.** Nutrient status impacts multilevel selection dynamics of ΔmtDNA. (A–B) Sub-organismal shift in ΔmtDNA frequency per generation, similar to *Figure 2B*, in wildtype (A) or null *daf-16*(mu86) (B) host genotype, on restricted or control diet. Adults were lysed at day 2 of adulthood, the same age at which the parents were lysed, to ensure that parents and their adult progeny were age-matched. Regressions compared using analysis of covariance. (C) Fraction of ΔmtDNA-carrying individuals at generation 8 of the competition experiment shown in (D) and *Figure 6—figure supplement 1A*, normalized to control-diet lines. Two-tailed Welch's t-test. (D) Organismal selection against ΔmtDNA as measured by population-wide ΔmtDNA frequency relative to average sub-organismal (heteroplasmic) frequency, similar to *Figure 2E*, in competing lineages of wildtype nuclear background, maintained on restricted or control diet. To isolate the change in ΔmtDNA frequency that occurs strictly due to organismal selection, we controlled for the confounding influence of sub-organismal ΔmtDNA dynamics by normalizing overall ΔmtDNA across each population to that of the non-competed lines at each generation. Because all individuals within the non-competed lines contain ΔmtDNA, the frequency across a non-competing population is equal to the average sub-organismal ΔmtDNA. Hence, normalizing ΔmtDNA to the non-competing lines accounts for sub-organismal ΔmtDNA dynamics and reveals the decline in ΔmtDNA that occurs strictly due to selection at the level of organismal fitness. Solid lines reflect mean normalized ΔmtDNA frequency. Non-competed lines not shown for visual simplicity. Linear regression analysis. (E) Fraction of ΔmtDNA-carrying individuals with *daf-16*(mu86) nuclear background at generation 8 of the competition experiment shown in (F) and *Figure 6—figure supplement 1B*, normalized to control-diet lines. Two-tailed Welch's t-test. (F) Organismal selection against ΔmtDNA as measured by population-wide ΔmtDNA frequency relative to average sub-organismal (heteroplasmic) frequency, similar to *Figure 2E* and (D), in competing lineages of *daf-16*(mu86) nuclear background, maintained on restricted or control diet. The ΔmtDNA frequency of each line, at each generation, was normalized to that of the non-competed lines in order to control for the confounding influence of sub-organismal ΔmtDNA dynamics, as was done in (D). Solid lines reflect mean normalized ΔmtDNA frequency. Non-competed lines not shown for visual simplicity. Linear regression analysis. (G) Schematic showing the influence of FoxO/DAF-16 on

*Figure 6 continued on next page*

*Figure 6 continued*

organismal selection against ΔmtDNA. Specifically, FoxO/DAF-16 protects ΔmtDNA from greater organismal selection during nutrient stress. (H) ΔmtDNA frequency, normalized to starting frequency, in non-competing lineages from the organismal competition experiment shown in (D) and (F). For comparison, dotted brown line represents data of non-competing data from the competition experiment on live OP50 *E. coli* (*Figure 2—figure supplement 1B*, gray lines). Linear regression analyses with Bonferroni correction for multiple comparisons. All experiments featured in this figure used nematodes maintained on a diet of UV-killed OP50 *E. coli* at 20 °C. Shaded regions show 95% C.I.

The online version of this article includes the following source data and figure supplement(s) for figure 6:

**Source data 1.** Summary statistics for frequency-dependent change in ΔmtDNA frequency at sub-organismal level, by diet and host genotype.
**Source data 2.** Summary statistics for frequency-dependent change in ΔmtDNA frequency at organismal level, by diet and host genotype.
**Figure supplement 1.** Quantification of ΔmtDNA frequency in competition experiments to measure organismal selection under altered dietary conditions and *daf-16* genotype.

overcome these challenges, we developed an approach to empirically quantify selection for cheater mtDNA at the sub-organismal (within-host) level by tracking cheater frequency within isolated parent-progeny lineages. At the organismal (between-host) level, we devised competition experiments enabling us to identify and measure the change in population-wide cheater frequency that occurs strictly due to the cost that the cheater imposes on host fitness. Our methodology not only makes it possible to empirically measure selection at different levels, but it also provides a powerful experimental approach that can be applied broadly to future studies seeking mechanistic insight on cooperator-cheater dynamics in hierarchically structured populations.

At the sub-organismal level, we note two trends describing the dynamics of the cheater genome ΔmtDNA. First, ΔmtDNA frequency declines between parent and embryo (*Figures 1C* and *2A*). This suggests germline purifying selection against deleterious mtDNA, a phenomenon observed across many species (*Ahier et al., 2018*; *Fan et al., 2008*; *Hill et al., 2014*; *Lieber et al., 2019*; *Ma et al., 2014*; *Stewart et al., 2008*). Although the molecular basis for germline purifying selection against ΔmtDNA is unknown, recent work in *Drosophila* has shown that mitochondrial protein synthesis in oocytes is localized around healthy mitochondria, providing a selection advantage for genomes that lack deleterious mutations (*Zhang et al., 2019*). Intriguingly, the same group also recently found

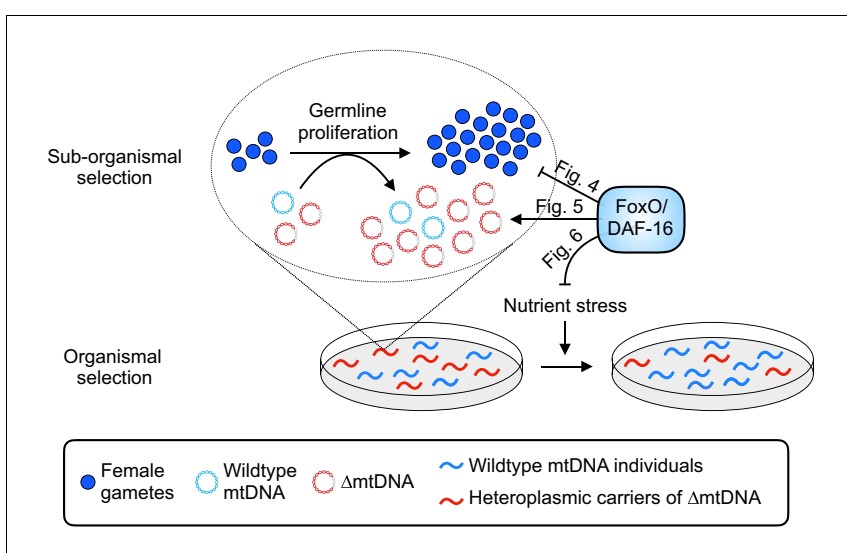

**Figure 7.** Summary of influence of diet and nutrient stress tolerance on multilevel selection dynamics of ΔmtDNA. At the sub-organismal level, FoxO/DAF-16 influences ΔmtDNA dynamics via two separate functions. On one hand, loss of insulin signaling results in activation of FoxO/DAF-16, which inhibits germline development (*Figure 4*). On the other hand, ΔmtDNA preferentially propagates by taking advantage of dietary nutrients but only when FoxO/DAF-16 is present (*Figure 5*), indicating that nutrient supply and FoxO/DAF-16-dependent nutrient sensing are each necessary, but not sufficient individually, for ΔmtDNA proliferation. During conditions of food scarcity, FoxO/DAF-16 partially shields ΔmtDNA from organismal selection (*Figure 6*), suggesting that nutrient supply and FoxO/DAF-16 promote ΔmtDNA propagation across organismal and sub-organismal selection levels.

that insulin signaling mediates purifying selection against a deleterious mtDNA variant in *Drosophila* eggs in a putatively FoxO-dependent manner (*Wang et al., 2019*), providing a potential basis by which maternal diet influences purifying selection between parent and embryo, as we observed (*Figure 3B, C*). Whether similar mechanisms underlie germline purifying selection against ΔmtDNA in *C. elegans* remains to be explored.

Following the initial decline from parent to embryo, ΔmtDNA proliferates across development in a frequency-dependent manner, a common feature of cheater entities (*Dobata and Tsuji, 2013*; *Dugatkin et al., 2005*; *Pruitt and Riechert, 2009*; *Riehl and Frederickson, 2016*; *Ross-Gillespie et al., 2007*). Specifically, the sub-organismal advantage of ΔmtDNA declines as its frequency approaches the range of 75–80% (*Figures 1C* and *2B*). One possible explanation for this observation involves resource availability. Previous work has suggested that ΔmtDNA copy number increases in addition to—not at the expense of—wildtype mtDNA (*Gitschlag et al., 2016*). In other words, as ΔmtDNA frequency increases, so does total mtDNA copy number. The apparent upper limit observed for sub-organismal ΔmtDNA proliferation could therefore reflect a depletion of resources required for genome replication. Another possibility is the activation of policing mechanisms, a common strategy for enforcing cooperation (*Özkaya et al., 2017*; *Riehl and Frederickson, 2016*). The host genome might prevent ΔmtDNA from rising beyond a certain level by increasing the targeted degradation of underperforming organelles via mitophagy. Consistent with this possibility, deletion of the mitophagy gene *pdr-1* was previously associated with an increase in ΔmtDNA frequency (*Gitschlag et al., 2016*; *Valenci et al., 2015*), suggesting that a policing mechanism encoded by the host genome limits the extent to which the cheater can proliferate.

In addition to ΔmtDNA frequency, we found temperature to be another condition that influences mitochondrial genome dynamics, with elevations in both ΔmtDNA frequency and overall mtDNA copy number at 25 °C (*Figure 3E,F*) compared to 16 °C (*Figure 3—figure supplement 1D,F*). Although the mechanistic basis for this temperature effect is unknown, one possible explanation involves the host response to stress. The presence of ΔmtDNA compromises mitochondrial function and elicits the activation of a physiological stress response that includes genes involved in mitochondrial biogenesis and protein quality control (*Gitschlag et al., 2016*; *Lin et al., 2016*). By seeking to restore mitochondrial function, the nuclear genome inadvertently promotes the propagation of ΔmtDNA in a vicious cycle (*Gitschlag et al., 2016*; *Lin et al., 2016*). Interestingly, more recent work has shown that warm temperature adversely affects mitochondrial function in adult nematodes (*Gaffney et al., 2018*), which raises the possibility that warm temperature could mimic the presence of ΔmtDNA in key ways that contribute to the propagation of the mutant genome. Future studies seeking to mechanistically characterize the impact of environmental stress as a modulator of ΔmtDNA proliferation may therefore benefit from considering temperature as a variable.

After measuring sub-organismal dynamics, we explored selection at the organismal level. Consistent with the predictions of multilevel selection, we quantitatively showed that selection at the sub-organismal level favors the cheater genome while selection at the organismal level favors the cooperative genome. These levels of selection appear to balance when ΔmtDNA frequency is near 60%, enabling ΔmtDNA to persist at this frequency across many generations (*Tsang and Lemire, 2002*). Having separately measured the effects of selection at different levels on a biological cheater, we turned to the question of how nutrient status influences selection at these different levels.

The role of nutrient status in shaping cooperator-cheater dynamics is not well understood. We first characterized the effect of maternal diet and nutrient sensing on sub-organismal cheater mtDNA dynamics. We found that nutrient abundance and insulin signaling promote mtDNA biogenesis in the germline, thereby providing the niche space for cheater proliferation. We further showed that the stress-response transcription factor FoxO/DAF-16 is necessary for the cheater to take advantage of nutrient supply and proliferate. Our findings reveal that while nutrient abundance may be necessary, it is not sufficient to promote cheater proliferation.

Interestingly, while we find that nutrient scarcity promotes the cooperative over cheater genotype, nutrient abundance is known to select for cooperation in other systems. We propose that the impact of nutrient availability depends on whether the cost or the benefit of cooperation is predominantly affected. Nutrient abundance can promote cooperation by reducing the cost of making a cooperative contribution (*Brockhurst et al., 2008*; *Connelly et al., 2017*; *Sexton and Schuster, 2017*). Alternatively, scarcity can increase the benefit of cooperating, a phenomenon we observe in heteroplasmy dynamics. Since the cheater genotype inflicts lower rates of respiration despite

increased overall mtDNA levels (*Tsang and Lemire, 2002*), the cooperative wildtype genotype achieves a greater bioenergetic payoff per nutrient invested. When nutrients are scarce, a sub-organismal shift occurs in favor of the more metabolically efficient wildtype genome. In conclusion, we find that the benefit of cooperation, as indicated by cooperator-biased replication, increases during nutrient scarcity and decreases during abundance.

In addition to nutrient abundance, DAF-16 is also required for sub-organismal cheater mtDNA proliferation. This could occur through compensatory biogenesis that favors underperforming organelles, inadvertently biasing replication toward the cheater genotype. Consistent with this possibility, FoxO/DAF-16 has been identified as a regulator of genes associated with mitochondrial biogenesis (*Tepper et al., 2013*; *Webb et al., 2016*). Alternatively, DAF-16 might passively permit cheater proliferation by alleviating stress. DAF-16 up-regulates the expression of multiple genes involved in energy metabolism and antioxidant defense (*Depuydt et al., 2014*; *Tepper et al., 2013*; *Webb et al., 2016*). By seeking to rescue ATP synthesis and protect against cellular damage, DAF-16 may relax the sub-organismal selective pressure to maintain optimal mitochondrial function, thereby permitting the spread of deleterious mtDNA mutants.

How does resource availability affect selection on cooperators and cheaters at the level of competing groups? Note that the female germline harbors the mtDNA molecules that compete for transmission. Selection on mtDNA genotype at the organismal level can thus be viewed as a group-level phenomenon, while sub-organismal selection represents the within-group level. On one hand, if resource scarcity selects for cooperation, groups with a higher proportion of cooperators should gain an extra fitness advantage over other groups during times of scarcity. On the other hand, exposure to cheating can lead to an evolutionary arms race whereby cooperators acquire resistance to cheaters, a phenomenon observed in bacteria and social amoebae (*Hollis, 2012*; *O'Brien et al., 2017*; *Khare et al., 2009*). Could food scarcity select for adaptations that reduce the impact of cheaters on group fitness? We propose that DAF-16 functions as an example of this type of stress tolerance. Although diet restriction compromised the sub-organismal advantage of ΔmtDNA, it had no effect on the organismal disadvantage, provided DAF-16 is present. However, in *daf-16* mutants, diet restriction intensified organismal selection against ΔmtDNA. We conclude that FoxO/DAF-16, known to prolong organismal survival during nutrient deprivation (*Greer et al., 2007*; *Hibshman et al., 2017*; *Kramer et al., 2008*), prevents food scarcity from subjecting the cheater to stronger organismal selection. Broadly, our findings suggest that the ability to cope with scarcity can promote group-level tolerance to cheating, inadvertently prolonging cheater persistence.

# Materials and methods

## Key resources table

| Reagent type (species) or resource | Designation | Source or reference | Identifiers | Additional information |
|---|---|---|---|---|
| Gene (*Caenorhabditis elegans*) | daf-2 | WormBase | Y55D5A.5 | |
| Gene (*Caenorhabditis elegans*) | daf-16 | WormBase | R13H8.1 | |
| Gene (*Caenorhabditis elegans*) | pdr-1 | WormBase | K08E3.7 | |
| Gene (*Caenorhabditis elegans*) | ced-3 | WormBase | C48D1.2 | |
| Gene (*Caenorhabditis elegans*) | drp-1 | WormBase | T12E12.4 | |

*Continued on next page*

*Continued*

| Reagent type (species) or resource | Designation | Source or reference | Identifiers | Additional information |
|---|---|---|---|---|
| Genetic reagent (*Caenorhabditis elegans*) | *him-8*(e1489); ΔmtDNA(*uaDf5*)/+ | Caenorhabditis Genetics Center | RRID:WB-STRAIN: WBStrain00024106 | LB138 |
| Genetic reagent (*Caenorhabditis elegans*) | *daf-2*(e1370) | Caenorhabditis Genetics Center | RRID:WB-STRAIN: WBStrain00004309 | CB1370 |
| Genetic reagent (*Caenorhabditis elegans*) | *daf-16*(mu86) | Caenorhabditis Genetics Center | RRID:WB-STRAIN: WBStrain00004840 | CF1038 |
| Genetic reagent (*Caenorhabditis elegans*) | *daf-2*(e1370); *daf-16*(mu86); muEx268 [*ges-1*p::GFP: *daf-16*(cDNA) + *odr-1*::RFP] | Caenorhabditis Genetics Center | RRID:WB-STRAIN: WBStrain00004876 | CF1827 |
| Genetic reagent (*Caenorhabditis elegans*) | *pdr-1*(gk448) | International *C. elegans* Gene Knockout Consortium | RRID:WB-STRAIN: WBStrain00036256 | VC1024 |
| Genetic reagent (*Caenorhabditis elegans*) | *ced-3*(ok2734) | International *C. elegans* Gene Knockout Consortium | RRID:WB-STRAIN: WBStrain00032755 | RB2071 |
| Genetic reagent (*Caenorhabditis elegans*) | *drp-1*(tm1108) | Shohei Mitani | RRID:WB-STRAIN: WBStrain00005196 | CU6372 |
| Genetic reagent (*Caenorhabditis elegans*) | bcIs39 V [*lim-7*p:: *ced-1*::GFP + *lin-15*(+)] | Barbara Conradt | RRID:WB-STRAIN: WBStrain00026469 | MD701 |
| Genetic reagent (*Caenorhabditis elegans*) | *tomm-20*::mCherry | Sasha de Henau | | TBDL58 |
| Genetic reagent (*Caenorhabditis elegans*) | ΔmtDNA(*uaDf5*)/+ in Bristol nuclear background | This study | See Materials and methods: Genetic crosses and genotyping |
| Genetic reagent (*Caenorhabditis elegans*) | *daf-2*(e1370); ΔmtDNA(*uaDf5*)/+ | This study | See Materials and methods: Genetic crosses and genotyping |
| Genetic reagent (*Caenorhabditis elegans*) | *daf-2*(e1370); *daf-16*(mu86); ΔmtDNA(*uaDf5*)/+ | This study | See Materials and methods: Genetic crosses and genotyping |
| Genetic reagent (*Caenorhabditis elegans*) | *daf-16*(mu86); ΔmtDNA(*uaDf5*)/+ | This study | See Materials and methods: Genetic crosses and genotyping |
| Genetic reagent (*Caenorhabditis elegans*) | *daf-2*(e1370); *pdr-1*(gk448) | This study | See Materials and methods: Genetic crosses and genotyping |
| Genetic reagent (*Caenorhabditis elegans*) | *daf-2*(e1370); *daf-16*(mu86); *pdr-1*(gk448) | This study | See Materials and methods: Genetic crosses and genotyping |

*Continued on next page*

*Continued*

| Reagent type (species) or resource | Designation | Source or reference | Identifiers | Additional information |
|---|---|---|---|---|
| Genetic reagent (*Caenorhabditis elegans*) | *daf-2*(e1370); *ced-3*(ok2734) | This study | | See Materials and methods: Genetic crosses and genotyping |
| Genetic reagent (*Caenorhabditis elegans*) | *daf-2*(e1370); *daf-16*(mu86); *ced-3*(ok2734) | This study | | See Materials and methods: Genetic crosses and genotyping |
| Genetic reagent (*Caenorhabditis elegans*) | *daf-2*(e1370); *drp-1*(tm1108) | This study | | See Materials and methods: Genetic crosses and genotyping |
| Genetic reagent (*Caenorhabditis elegans*) | *daf-2*(e1370); *daf-16*(mu86); *drp-1*(tm1108) | This study | | See Materials and methods: Genetic crosses and genotyping |
| Genetic reagent (*Caenorhabditis elegans*) | *daf-2*(e1370); *tomm-20*::mCherry | This study | | See Materials and methods: Genetic crosses and genotyping |
| Genetic reagent (*Caenorhabditis elegans*) | *daf-2*(e1370); *daf-16*(mu86); *tomm-20*::mCherry | This study | | See Materials and methods: Genetic crosses and genotyping |
| Genetic reagent (*Caenorhabditis elegans*) | *daf-16*(mu86); *tomm-20*::mCherry | This study | | See Materials and methods: Genetic crosses and genotyping |
| Genetic reagent (*Caenorhabditis elegans*) | *daf-2*(e1370); bcIs39 V [*lim-7*p::*ced-1*::GFP + *lin-15*(+)] | This study | | See Materials and methods: Genetic crosses and genotyping |
| Genetic reagent (*Escherichia coli*) | HT115 strain expressing Y55D5A_391.b (*daf-2*) ORF plasmid clone | Ahringer Group, Source BioScience | 3318_Cel_RNAi_complete | See Materials and methods: Knockdown of gene expression |
| Sequence-based reagent | PCR primers used in this study | This study | | See Materials and methods: Genetic crosses and genotyping; Quantification of mtDNA copy number and ΔmtDNA frequency |
| Peptide, recombinant protein | BlpI restriction endonuclease | New England Biolabs | Cat#R0585L | See Materials and methods: Genetic crosses and genotyping |
| Chemical compound, drug | Isopropyl-β-D-thioga lactopyranoside | Research Products International | Cat#I56000-1 | See Materials and methods: Genetic crosses and genotyping |
| Chemical compound, drug | 4′,6-diamidino-2-phenylindole (DAPI) | Thermo Fisher Scientific | Cat#D1306 | |
| Chemical compound, drug | Paraformaldehyde | Electron Microscopy Sciences | Cat#15710 | |
| Chemical compound, drug | Levamisole | Fisher Scientific | Cat#AC187870100 | |
| Commercial assay, kit | DreamTaq Green DNA Polymerase | Thermo Fisher Scientific | Cat#EP0713 | |
| Commercial assay, kit | Seahorse XFe96 FluxPak | Agilent | Cat#102601–100 | |

*Continued on next page*

*Continued*

| Reagent type (species) or resource | Designation | Source or reference | Identifiers | Additional information |
|---|---|---|---|---|
| Other | Eppendorf 96-well twin.tec PCR plates | Fisher Scientific | Cat#951020303 | |
| Other | QX200 ddPCR Eva Green Supermix | Bio-Rad | Cat#1864034 | |
| Other | Automated Droplet Generation Oil for EvaGreen | Bio-Rad | Cat#1864112 | |
| Other | DG32 Automated Droplet Generator Cartridges | Bio-Rad | Cat#1864108 | |
| Other | Droplet Reader Oil for ddPCR | Bio-Rad | Cat#1863004 | |
| Software, algorithm | QuantaSoft | Bio-Rad | Cat#1864011 | |
| Software, algorithm | Zen | Carl Zeiss Microscopy GmbH | RRID:SCR_013672 | |
| Software, algorithm | Prism eight for macOS | GraphPad Software, Inc | RRID:SCR_002798 | Version 8.1.2 |
| Software, algorithm | ImageJ | Wayne Rasband, NIH | RRID:SCR_003070 | Version 1.49 |

## Nematode culture

*C. elegans* strains used in this study were maintained on 60 mm standard nematode growth medium (NGM) plates seeded with live OP50 *E. coli* bacteria as a food source, unless otherwise indicated below. Nematode strains were incubated at 20 ˚C unless otherwise indicated. Age-matched nematodes were used in all experiments with the exception of the multigenerational competition experiment (see below).

## Nematode lysis

To prepare nematodes for genotyping and quantification of mtDNA copy number and ΔmtDNA frequency, nematodes were lysed using the following protocol. Nematodes were transferred to sterile PCR tubes or 96-well PCR plates containing lysis buffer with 100 µg/mL proteinase K. Lysis buffer contained 50 mM KCL, 10 mM Tris pH 8.3, 2.5 mM $MgCl_2$, 0.45% Tween 20, 0.45% NP-40 (IGEPAL), and 0.01% gelatin, in deionized $H_2O$. Volume of lysis buffer varied by worm count: 10 µL for individual adults, pooled larvae, or pooled embryos; 20 µL for 5 or 10 pooled adults; 50 µL for pooled nematodes of mixed age (competition experiments, see below). Each tube or plate was then incubated at −80 ˚C for 10 min, then at 60 ˚C for 60 min (90 min for pooled nematodes), and then at 95 ˚C for 15 min to inactivate the proteinase K. Nematode lysates were then stored at −20 ˚C.

## Genetic crosses and genotyping

To control for nuclear effects on ΔmtDNA proliferation, hermaphroditic nematodes carrying the ΔmtDNA allele *uaDf5* were serially back-crossed into a male stock of the Bristol (N2) *C. elegans* nuclear background for six generations. To investigate the role of insulin signaling in selfish mitochondrial genome dynamics, the alleles *daf-2*(e1370) and *daf-16*(mu86) were introduced to the ΔmtDNA heteroplasmic lineage by classical genetic crosses. To investigate the mechanistic basis by which the insulin signaling pathway regulates mtDNA levels, mutant alleles affecting various putative downstream processes were genetically crossed into the insulin signaling-defective nuclear genotypes. Specifically, the parkin-dependent mitophagy-defective *pdr-1*(gk448), the mitochondrial fission-defective *drp-1*(tm1108), and the apoptosis-defective *ced-3*(ok2734) were each genetically combined with *daf-2*(e1370), both with and without the *daf-16*(mu86) allele. Nuclear genotype was confirmed by PCR using the following oligonucleotide primers:

Mutant and wildtype mtDNA:

Exterior forward: 5'-CCATCCGTGCTAGAAGACAA-3'
Interior forward: 5'-TTGGTGTTACAGGGGCAACA-3'

Reverse: 5'-CTTCTACAGTGCATTGACCTAGTC-3'
daf-2
Forward: 5'-CATCAAGATCCAGTGCTTCTGAATCGTC-3'
Reverse: 5'-CGGGATGAGACTGTCAAGATTGGAG-3'
daf-16
Forward: 5'-CACCACGACGCAACACACTAATAGTG-3'
Exterior reverse: 5'-CACGAGACGACGATCCAGGAATCG-3'
Interior reverse: 5'-GGTCTAAACGGAGCAAGTGGTTACTG-3'
pdr-1
Exterior forward: 5'-GAATCATGTTGAAAATGTGACGCGAG-3'
Interior forward: 5'-CTGACACCTGCAACgtaggtcaag-3'
Reverse: 5'-GATTTGACTAGAACAGAGGTTGACGAG-3'
drp-1
Forward: 5'-CGTCGGATCACAGTCGGC-3'
Reverse: 5'-GCACTGACCGCTCTTTCTCC-3'
ced-3
Exterior forward: 5'-cagtactccttaaaggcgcacacc-3'
Interior forward: 5'-gattggtcgcagttttcagtttagaggg-3'
Reverse: 5'-CGATCCCTGTGATGTCTGAAATCCAC-3'

The insulin signaling receptor allele *daf-2*(e1370) introduces a point mutation that eliminates a *BlpI* restriction endonuclease recognition site. Following PCR amplification, *daf-2* PCR products were incubated with *BlpI* and New England BioLabs CutSmart buffer at 37 °C for 2 hr prior to gel electrophoresis. Fluorescent reporters used in this study were genotyped by fluorescence microscopy.

## Quantification of mtDNA copy number and ΔmtDNA frequency

Quantification of mtDNA copy number and ΔmtDNA frequency was accomplished using droplet digital PCR (ddPCR). Nematodes were lysed as described above. Lysates were then diluted in nuclease-free water, with a dilution factor varying depending on nematode concentration: 20x for embryos, 200x for pooled larvae, 200x for single adults, 1000x for pooled adults, 20,000x for pooled nematodes of mixed age from the competition experiments (control diet) or 2000x for pooled nematodes of mixed age from the competition experiments (restricted diet). The lower dilution factor for the lysates collected from the restricted diet condition was due to the smaller population sizes of nematodes raised on a restricted diet, which arises from reduced fecundity under diet restriction and was reflected in the number of nematodes present in these lysates. Next, either 2 μL or 5 μL of each dilute nematode lysate was combined with 0.25 μL of a 10 μM aliquot of each of the following oligonucleotide primers:

For quantifying wildtype mtDNA:

5'-GTCCTTGTGGAATGGTTGAATTTAC-3'
5'-GTACTTAATCACGCTACAGCAGC-3'

For quantifying ΔmtDNA:

5-'CCATCCGTGCTAGAAGACAAAG-3'
5-'CTACAGTGCATTGACCTAGTCATC-3'

Mixtures of dilute nematode lysate and primer were combined with nuclease-free water and Bio-Rad QX200 ddPCR EvaGreen Supermix to a volume of 25 μL in Eppendorf 96-well twin.tec PCR plates. Droplet generation and PCR amplification were performed according to manufacturer protocol with an annealing temperature of 58 °C. For amplification of heteroplasmic nematode lysates, wildtype and ΔmtDNA primers were combined in the same reaction, and each droplet was scored as containing either wildtype or mutant template using the 2D amplitude (dual-wavelength) clustering plot option in the Bio-Rad QuantaSoft program.

## Respiration assay

Basal and maximum oxygen consumption rates were measured using the Seahorse XFe96 Analyzer in the High Throughput Screening Facility at Vanderbilt University. One day before experimentation, each well of a 96-well sensor cartridge that comes as part of the Seahorse XFe96 FluxPak was

incubated with 200 µL of the Seahorse XF Calibrant Solution. On the day of the experiment, 10–20 L4-stage animals were randomly sampled from either a stock population stably maintaining ΔmtDNA in the range of 50–80% (population mean approximately 60%), or from a wildtype control (Bristol strain). The animals were placed into each well of the cell culture microplate. Wells contained either M9 buffer or 10 µM FCCP. After calibration, 16 measurements were performed at room temperature. Measurements 12 through 16 were averaged and normalized to number of worms per well.

### Fertility

To assay fertility, day-2 adult nematodes were individually transferred onto NGM plates seeded with live OP50 *E. coli* and incubated at 20 °C for 4 hr. The adults were then individually lysed as described above. Fertility was scored as the average number of viable progeny produced per hour during the 4 hr window, where viable progeny were identified as those that had progressed from embryos to larvae within 24 hr of being laid. The ΔmtDNA frequency of each parent was determined using ddPCR as described above. We assayed 48 individual ΔmtDNA-containing parents for the purpose of correlating ΔmtDNA frequency with fecundity using a linear regression. However, the regression was not significant. We then collapsed the heteroplasmic data points into two bins corresponding to low and high ΔmtDNA frequency (below and above the population mean of 60%, respectively) in order to compare their corresponding fecundities to that of a wildtype control. We set the sample size of the high-ΔmtDNA frequency bin from N = 35 to N = 12, to match the sample size of the low-ΔmtDNA frequency bin. To accomplish this, 12 samples were randomly selected to be retained and 23 samples were randomly selected to be discarded. To confirm that this did not affect our statistical analysis, this was repeated three times and statistical analysis was performed on each data set containing a randomly-selected sample of N = 12 for the high-ΔmtDNA frequency bin. In each case, the presence of ΔmtDNA at >60% frequency continued to correspond to a significantly lower fertility rate than wildtype controls.

### Development

The impact of ΔmtDNA levels on development was assayed by comparing ΔmtDNA frequency with developmental stage for each nematode in a population of age-synchronized larvae. To age-synchronize larvae, multiple mature heteroplasmic adults carrying ΔmtDNA in the Bristol nuclear background were transferred to an NGM plate seeded with live OP50 *E. coli* and allowed to lay eggs at 20 °C for 2 hr. Adults were then removed from the plate. After 48 hr, each nematode was individually lysed and its respective larval stage (L2, L3, or L4) was annotated. None of the nematodes had yet reached adulthood at this point. Embryos that failed to transition to larvae were discarded. The ΔmtDNA frequency of each larval nematode was determined using ddPCR as described above.

### Sub-organismal selection assay

Sub-organismal selection for ΔmtDNA was quantified by measuring changes in ΔmtDNA frequency as a function of developmental stage, and as a function of initial (parental) ΔmtDNA frequency, within a single generation. This was accomplished using two complementary approaches. In the first approach, three individual age-synchronized parents were selected according to initial ΔmtDNA frequency (parents with low, middle, and high frequency). One age-matched (L4-stage) nematode was picked at random under a dissecting microscope from each line respectively maintained under artificial selection for low (<50%), medium (50–70%), and high (>70%) ΔmtDNA frequency. Each of these nematodes was placed onto a fresh NGM plate seeded with live OP50 *E. coli* and incubated for 2 days at 20 °C. Each day-2 adult was then transferred to a fresh food plate every 4 hr and allowed to lay embryos. At each 4 hr time point, approximately one third of the embryos produced were individually lysed. After 12 hr, the adults were individually lysed. A 12 hr time window for embryo production was chosen in order to generate a sufficient offspring count to allow for the establishment of single-brood frequency distributions of ΔmtDNA. The 12 hr time window was divided into 4 hr segments in order to maintain age-synchronicity, as each larva was lysed within 4 hr of being laid across the entire 12 hr period. After 2 days at 20 °C, approximately one third of the L4-stage larvae were individually lysed in the same 4 hr segments to maintain age synchronicity. After an additional 2 days at 20 °C, the remaining one third of offspring were individually lysed in 4 hr segments, as they reached the same age at which their respective parent was lysed. The ΔmtDNA frequency of each

individual was determined using ddPCR as described above and a ΔmtDNA frequency distribution was generated for each offspring life stage.

In the second approach, multiple L4-stage heteroplasmic nematodes were picked at random under a dissecting microscope from the stock of nematodes carrying ΔmtDNA in the Bristol nuclear background. These larvae were transferred to a fresh food plate and incubated for 2 days at 20 ℃. The day-2 adults were then segregated onto individual plates and incubated for 4 hr at 20 ℃ to produce age-synchronized progeny. After 4 hr, each parent was individually lysed. Three embryos from each parent were also lysed at the same time, in one pooled lysate per three same-parent embryos. After 2 days, three L4-stage larvae were pooled and lysed from each parent, similar to the lysis of embryos. After another 2 days, three adult progeny were pooled and lysed from each parent as they reached the age at which the parents were lysed. Each parent-progeny lineages was individually segregated from the rest. Since ΔmtDNA impacts fecundity, the progeny from parents on the lower end of the ΔmtDNA frequency are expected to be overrepresented in the offspring sampled from a mixed cohort of parents. Lineages were therefore segregated to ensure that the ΔmtDNA frequency from each progeny lysate was being compared with that of its own respective parent, in order to minimize the effect of organism-level selection on ΔmtDNA. In addition, progeny from each timepoint were lysed in pools of three to reduce the effect of random drift on ΔmtDNA frequency. The ΔmtDNA frequency of parents and each developmental stage of progeny was determined using ddPCR as described above. For the measurement of sub-organismal selection on ΔmtDNA under nutrient-variable conditions, each parent was raised from embryo to adult under its respective dietary condition (diet restriction or control).

## Experimental evolution (organismal selection)

Selection against ΔmtDNA that occurs strictly at the level of organismal fitness was measured using a competition assay. Heteroplasmic nematodes carrying ΔmtDNA in the Bristol nuclear background were combined with Bristol-strain nematodes on 10 cm NGM plates seeded with live OP50 *E. coli*. For the first generation, heteroplasmic and Bristol strain nematodes were age-synchronized. Age synchronization was accomplished using a bleaching protocol. Nematodes from a mixed-age stock food plate were washed off the plate and into a sterile 1.7 mL microcentrifuge tube with nuclease-free water. The water was brought to a volume of 750 μL. The volume of each tube was brought to 1 mL by adding 100 μL of 5 N NaOH and 150 μL of 6% bleach. Each nematode tube was incubated at room temperature for 10 min with light vortexing every 2 min to rupture gravid adults and release embryos. Nematode tubes were centrifuged for 1 min at 1000x g to pellet the nematode embryos. To wash the nematode pellets, the supernatant was removed and replaced with 1 mL of nuclease-free water. After a second spin for 1 min at 1000x g, the water was removed and the nematode embryos were resuspended in 100 μL M9 buffer. The resuspended embryos were then transferred to glass test tubes containing 500 μL M9 buffer and incubated overnight at room temperature on a gentle shaker to allow hatching and developmental arrest at the L1 larval stage. On the following day, a glass Pasteur pipette was used to transfer approximately equal quantities of heteroplasmic and homoplasmic-wildtype nematodes onto the 10 cm food plates. Approximately 500 nematodes were transferred to each plate. In addition to eight competition lines, eight control lines were established by transferring only heteroplasmic nematodes randomly selected from the same overnight incubation tubes onto food plates, with no homoplasmic-wildtype nematodes to compete against.

Every 3 days, the generation for each experimental line was reset. To do this, nematodes were washed off the plates using sterile M9 buffer into a sterile 1.7 mL collection tube. Approximately 500 nematodes of mixed age from each line were transferred to a fresh food plate. An additional 500 nematodes were lysed together in a single pooled lysate. Finally, 48 additional adults from each competition line were lysed individually in order to determine the fraction of heteroplasmic nematodes in each competition line at each generational time point. This experiment was continued for 10 consecutive generations.

Experimental evolution was also carried out to quantify nutrient-conditional organism-level selection. These conditions included 10 cm NGM plates seeded with a restricted or a control diet consisting of UV-killed OP50 *E. coli* (prepared as described below). Two iterations of this experiment were conducted, one with wildtype nuclear genotype and one with nematodes homozygous for the null *daf-16*(mu86) allele. Due to the smaller brood sizes among nematodes raised on a restricted diet, 200 nematodes were transferred and another 200 lysed at each generation, instead of the 500 as in

the case of the experiment using a live bacterial diet. For these nutrient-conditional competition experiments, six replicate lines were propagated for each condition for a total of 8 consecutive generations. Lysis and quantification of ΔmtDNA frequency by ddPCR were performed as described above.

### Diet restriction

Diet restriction was accomplished using variable dilutions of UV-inactivated OP50 *E. coli* bacterial lawns on NGM plates. To prepare diet-restricted food plates, 1 L of sterile 2xYT liquid microbial growth medium was inoculated with 1 mL of live OP50 *E. coli* (suspended in liquid LB) using a sterile serological pipette. The inoculated culture was then incubated overnight on a shaker at 37 ˚C. The following day, the OP50 *E. coli* was pelleted by centrifugation for 6 min at 3,900 rpm. The pellet was resuspended to a bacterial concentration of approximately $2 \times 10^{10}$ cells/mL in sterile M9 buffer. This suspension was seeded onto NGM plates (control) or further diluted 100-fold to $2 \times 10^8$ cells/mL in sterile M9 buffer before being seeded onto NGM plates (diet restriction). Plates were incubated upright at room temperature 4 hr to allow the lawns to dry. To inhibit bacterial growth, plates were irradiated with UV radiation using a Stratagene UV Stratalinker 1800 set to $9.999 \times 10^5$ μJ/cm$^2$. To confirm inhibition of bacterial growth, UV-treated plates were incubated overnight at 37 ˚C. Animals were picked at random under a dissecting microscope onto either control or diet restriction plates.

### Insulin signaling inactivation

Insulin signaling was conditionally inactivated using the allele *daf-2*(e1370), encoding a temperature-sensitive variant of the *C. elegans* insulin receptor homolog. Because complete loss of insulin signaling during early larval development results in a stage of developmental arrest (dauer), age-synchronized nematodes were incubated at the permissive temperature of 16 ˚C until reaching the fourth and final larval stage. L4-stage larvae were then picked at random under a dissecting microscope for either transfer to the restrictive temperature of 25 ˚C or for continued incubation at 16 ˚C as a control. After 4 days of incubation, mature adults were lysed and ddPCR quantification of ΔmtDNA frequency was performed as described above. To follow up on the downstream mechanism by which insulin signaling regulates mtDNA dynamics, homoplasmic nematodes were incubated at the restrictive temperature of 25 ˚C and mtDNA copy number was measured using the same ddPCR primer pair that was used for quantifying the wildtype mtDNA in heteroplasmic worms.

### Knockdown of gene expression

Expression knockdown of the *C. elegans* insulin signaling receptor homolog, *daf-2*, was accomplished using feeder plates. Cultures consisting of 2 mL LB and 10 μL ampicillin were inoculated with a bacterial culture obtained from Source BioScience harboring the Y55D5A_391.b (*daf-2*) ORF plasmid clone and incubated overnight on a shaker at 37 ˚C. Bacteria containing the empty plasmid vector were used to establish a control diet. The following day, 750 μL of culture was transferred to a flask containing 75 mL LB and 375 μL ampicillin and incubated 4–6 hr on a shaker at 37 ˚C, until $OD_{550-600}$ >0.8. An additional 75 mL LB was added to the culture along with another 375 μL ampicillin and 600 μL 1 M isopropyl β-D-1-thiogalactopyranoside (IPTG) to induce expression of the small interfering RNA. Cultures were incubated another 4 hr on a shaker at 37 ˚C. Cultures were then centrifuged for 6 min at 3,900 rpm and the resulting bacterial pellets were each resuspended in 6 mL M9 buffer with 8 mM IPTG. Next, 250 μL of resuspension was seeded onto each NGM plate. Plates were allowed to dry at room temperature in the dark and then stored at 4 ˚C until use. Synchronized L4-stage nematodes were picked at random under a dissecting microscope onto either RNAi knockdown or control plates and incubated at 25 ˚C until day 4 of adulthood to match the conditions that were used for the *daf-2* mutant allele. Day-4 adults were lysed and their mtDNA copy number was quantified using ddPCR as described above.

### Live imaging

Overall mitochondrial content across the wildtype and defective insulin signaling genotypes was measured using the mitochondrial reporter TOMM-20::mCherry. Age-synchronized nematodes were incubated for 2 days from the L4 stage to mature adulthood at 25 ˚C, immobilized with 10 mM

levamisole, and placed on the center of a 2% agarose pad on a microscope slide. Nematodes were imaged at 10x magnification using a Leica DM6000 B compound fluorescence microscope and mitochondrial fluorescence was quantified using ImageJ. Apoptosis was imaged in *daf-2*(e1370) mutant nematodes and wildtype controls using the CED-1::GFP reporter. Age-synchronized nematodes were incubated for 2 days from the L4 stage to mature adulthood at 25 °C before being immobilized and mounted on microscope slides as described above. Apoptotic cells were imaged using a Zeiss LSM 880 Confocal Laser Scanning microscope at 20x magnification.

## Staining and imaging of germline nuclei

Nematode germline nuclei were quantified across age-synchronized mature adults homozygous for *daf-2*(e1370) or *daf-16*(mu86), as well as in double-mutants and wildtype controls. For each genotype, age-synchronized L4-stage nematodes were incubated for 2 days at 25 °C and then placed in a plate containing 3 mL of PBS with 200 µM levamisole. To dissect the nematode gonads, each adult was decapitated using two 25G × 1' hypodermic needles in a scissor-motion under a dissecting microscope. Dissected gonads were fixed for 20 min in 3% paraformaldehyde. Fixed gonads were transferred to a glass test tube using a glass Pasteur pipette and the paraformaldehyde was replaced with PBT (PBS buffer with 0.1% Tween 20) and incubated for 15 min at room temperature. The PBT was then replaced with PBT containing 100 ng/mL 4',6'-diamidino-2-phenylindole dihydrochloride (DAPI) and the gonads were incubated in darkness for another 15 min at room temperature. Gonads were then subjected to 3x consecutive washes, each consisting of a 1 min centrifugation at 1,000 rpm followed by replacement of the PBT. Gonads were then mounted directly onto a 2% agarose pad on the center of a microscope slide and imaged using a Zeiss LSM 880 Confocal Laser Scanning microscope at 20x magnification.

## Statistical analysis

The effect of initial ΔmtDNA frequency on sub-organismal ΔmtDNA dynamics was observed in three experiments, one of which included the variables of diet and host *daf-16* genotype. The effect of FoxO/DAF-16-dependent insulin signaling on mtDNA copy number was observed in five experiments using the temperature-sensitive *daf-2*(e1370) allele, plus once more using knockdown of *daf-2* gene expression. Effect of diet restriction on sub-organismal ΔmtDNA proliferation was observed in three experiments, two of which included the variable of host *daf-16* genotype. The effect of organismal selection on population-wide ΔmtDNA prevalence was observed in two separate competition experiments, one of which included the variables of diet and host *daf-16* genotype. Each data point represents a biological replicate. For each experiment, sample sizes and number of animals per sample are provided in the respective figure legend. Sample sizes were chosen based on prior qualitative assessment of the impact of conditions such as diet, temperature, host age and host genotype on ΔmtDNA frequency and mtDNA copy number. For each experiment, significance was determined using the statistical test indicated in the respective figure legend, along with the indicated multiple comparisons test whenever two or more groups are compared.

## Acknowledgements

We thank the members of the Patel Laboratory (James P Held, Cait S Kirby, Nikita Tsyba, Benjamin R Saunders, Cassidy A Johnson), Janet M Young, Mia T Levine, Sarah E Zanders, Harmit S Malik, and Antonis Rokas for their valuable feedback on the manuscript. Some strains were provided by the CGC, which is funded by NIH Office of Research Infrastructure Programs (P40 OD010440). This work was generously supported by R01 GM123260 (MRP), the Ruth L Kirschstein National Research Service Award Individual Predoctoral Fellowship 1F31GM125344 (BLG), and the Vanderbilt University Medical Center Diabetes Research and Training Center Pilot and Feasibility Grant. Confocal microscopy imaging was performed through the use of the Vanderbilt Cell Imaging Shared Resource (supported by NIH grants CA68485, DK20593, DK58404, DK59637 and EY08126). Quantification of mtDNA copy number and ΔmtDNA frequency was conducted with the help of the Simon A Mallal Laboratory at Vanderbilt University Medical Center.

## Additional information

### Funding

| Funder | Grant reference number | Author |
| --- | --- | --- |
| National Institute of General Medical Sciences | GM123260 | Maulik R Patel |
| National Institute of General Medical Sciences | 1F31GM125344 | Bryan L Gitschlag |
| The Vanderbilt Diabetes Research and Training Center | Pilot and Feasibility Grant | Bryan L Gitschlag Maulik R Patel |

The funders had no role in study design, data collection and interpretation, or the decision to submit the work for publication.

### Author contributions

Bryan L Gitschlag, Conceptualization, Data curation, Formal analysis, Funding acquisition, Validation, Investigation, Visualization, Methodology, Writing - original draft, Writing - review and editing; Ann T Tate, Formal analysis, Methodology, Writing - review and editing; Maulik R Patel, Conceptualization, Resources, Data curation, Formal analysis, Supervision, Funding acquisition, Validation, Methodology, Project administration, Writing - review and editing

### Author ORCIDs

Ann T Tate http://orcid.org/0000-0001-6601-0234
Maulik R Patel https://orcid.org/0000-0003-3749-0122

### Decision letter and Author response

Decision letter https://doi.org/10.7554/eLife.56686.sa1
Author response https://doi.org/10.7554/eLife.56686.sa2

## Additional files

### Supplementary files

- Transparent reporting form

### Data availability

All data generated or analyzed during this study are included in the manuscript and supporting files.

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
