## [Decision Letter]

**Acceptance summary:**

Using a heteroplasmic strain of *C. elegans*, authors examined the dynamics of "cheating" mitochondrial DNA within and across a generation, and how this dynamics was affected by nutrient status. This work is elegant and quantitative, and makes a timely contribution to multi-level selection (within-host selection and between-host selection) literature.

**Decision letter after peer review:**

Thank you for submitting your article "Selfish mitochondria exploit nutrient status to proliferate across the different levels of selection" for consideration by *eLife*. Your article has been reviewed by four peer reviewers, including Wenying Shou as the Reviewing Editor and Reviewer #1, and the evaluation has been overseen by George Perry as the Senior Editor. The following individuals involved in review of your submission have agreed to reveal their identity: M Florencia Camus (#2); Hansong Ma (Reviewer #3); Eric Haag (Reviewer #4).

The reviewers have discussed the reviews with one another and the Reviewing Editor has drafted this decision to help you prepare a revised submission.

This paper needs a serious revision in writing. Overall, we feel that although interesting, you oversimplified your work so much that it ended up being very difficult to follow. Temperature and nutrient are two variables in your experiments, yet you chose to focus on the latter without explaining why (even though the former clearly had an effect). The frequency-dependent advantage of cheating mtDNA over wild type mtDNA needs to be clearly stated and emphasized to reduce confusion. See more detailed comments below.

Given the interest in the science but critical need for major changes to the presentation, the full reviewer comments are provided below for you to consider as you rework the paper.

Summary:

Glitschlag et al. examine the fate of cheating mitochondria in *C. elegans* as a function of environmental (nutrient and temperature) status. Mitochondria DNA (mtDNA) with a large deletion (cheating mtDNA) has a fitness advantage over normal cooperative mtDNA during organismal development, although the frequency of cheating mtDNA drops somewhat from parent to embryo. Thus, cheating mtDNA has a within-host advantage over cooperating mtDNA, especially when cheating mtDNA is initially rare. On the other hand, host with high fraction of cheating mtDNA suffers fitness cost. Thus, cheating mtDNA has a between-host disadvantage compared to cooperating mtDNA. During germline development (when mtDNA undergoes most proliferation), if nutrients are in excess and temperature is upshifted, insulin signalling through DAF-2 inhibits DAF-16, which allows mtDNA to proliferate, which in turn allows cheating mtDNA to takeover. In this case, DAF-16 inhibits mtDNA proliferation (which deters takeover by cheating mtDNA). In normal temperature, DAF-16 promotes cheating mtDNA within host in both poor and rich diets. Thus, the role of DAF-16 is complex.

Reviewer #1:

The work is interesting, and techniques for quantifying within- and between-host fitness difference between cheating and cooperating mtDNA are nice. The paper does not read smoothly especially at the DAF-16/diet/temperature part. The part of the Price equation is not needed, and seems incorrect/incomplete anyways. Feel free to save the Price equation part which, after a more rigorous analysis, can stand on its own as an independent article.

1) The Abstract does not give a clear picture of what the story is. Contents of supplementary figures are rarely given in the main text. Figure legends are also very brief.

2) Figure 1: When cheater mtDNA gets to 80%, it can no longer increase in frequency. Why is that? For the general audience, it is also useful to add a bit more detail on why cheating mtDNA declines from parent to embryo.

3) Figure 2C: I am not sure what this means. Does this (decreasing or similar %cheating mtDNA over development) contradict your Figure 1 (increasing %cheating mtDNA over development)?

4) Figure 2E: I find the figure hard to digest. I would like to see three panels plotted in the same figure for comparison: (i) within host advantage of cheating mtDNA (Figure 1D); (ii) between host disadvantage of cheating mtDNA (Figure 2—figure supplement 1B); and (iii) population level mtDNA (Figure 2E). According to Price Equation, (iii) should be a function of (i) and (ii) and variations in cheating mtDNA among offspring embryos.

5) Price equation: In the Price equation, there are two terms (as suggested in Figure 3—figure supplement 1, although note the typo: () should be ()). The first term of covariance describes how the host fecundity covaries with cooperative mtDNA in the host (and is a function of variations in cheating mtDNA among offspring embryos), and the second term of expectation describes how the cooperative mtDNA is lost to cheating mtDNA within host. Either you have misunderstood Price equation, or your writing is very confusing, especially when you sum up the sub-organismal and organismal covariance terms.6) Subsection “Insulin signaling influences sub-organismal proliferation of ∆mtDNA”, last paragraph: DAF-16 promotes cheating mtDNA in both rich and poor diet, and not just rich diet, correct? (Figure 6B). In general, the two environmental variables (temperature and diet) plus the genotype variables makes data interpretation and writing (and reading) tricky. Your interpretation has mainly focused on diet despite the apparent importance of temperature (Figure 4F). For example, what happens during restrictive diet and temperature shift? I am not necessarily asking you to do more experiments, but careful writing and illustration is critical when biology gets complex.

7) Figure 7: This should ideally be the summary figure (data can go to the supplement?). You can have schematics for different environments and genotypes. In each schematic, you can include numbers such as selection coefficient of cheating mtDNA from parent to embryo (and variations among embryos), from embryo to Adult (within host advantage, which depends on initial frequency of cheating mtDNA), and at the organismal level (between-host disadvantage). Again, I am not asking you to do more experiments to fill all entries, but putting everything together makes it easier to spot patterns/discrepancies.

Reviewer #2:

Authors examine mitochondrial proliferation dynamics in a very nice *C. elegans* model system. Using a heteroplasmic strain of worm, they investigate the proportion of a deleterious "ΔmtDNA" passed onto the next generation. This ΔmtDNA lacks several core mitochondrial genes as well as tRNAs, and authors demonstrate that is impacts several physiological traits. Interestingly, in developing worms they see a decrease in the frequency of the deleterious haplotype, however as the worm matures this percentage is increased. When in competition, however, the detrimental effects of the ΔmtDNA are clear, with a decrease in frequency over generations. Authors then go on to examine this dynamic across two different environments; different nutritional environments and temperatures. They find that two insulin-signalling genes (DAF-2, DAF-16) are involved in mtDNA proliferation, but this is contingent on the temperature that worms were at.

While I really enjoyed this paper, authors could tone down their language a bit. Furthermore, bits of the paper could be re-written to give a clearer picture to the reader. This is a paper with a lot of technical experiments, and it would be good for the authors describe experimental setup, and statistical inferences in the Results in a clearer way. Personally, I think it is a very timely contribution to the scientific literature, and is not only important to the mitochondrial field, but across the wider biological sciences.

Comment about statistics:

1) Are the statistical tests performed in relative or raw datapoints? If they are performed on the relative data, can authors please justify this? I find that by standardising data, differences appear to be bigger and less biologically meaningful.

2) Along these lines, it would be good for the authors to tell the reader in the Results sections about these relative values (how they were calculated, and what they mean). There is a little bit in the figure legends, but this is of great importance for biological conclusions.

Other comments:

1) Subsection “An experimental strategy to isolate selection on a selfish mitochondrial genome at different levels”, second paragraph: I think you mean Figure 1C and D in this sentence instead of Figure 1D and E?

2) Figure 1C – It would be very helpful to write on each panel "low", "medium" or "high". I wrote it on my printed copy and it made the figure easier to interpret.

3) Figure 1D – I understand what you are trying to get to with this figure, but maybe a better way to present it would be using simple boxplots, or mean ± SE (within each of their respected categories). To me it looks a bit messy, and it took me a bit to get my head around it. I'm mainly saying this because this figure conveys a very important message, and the reader would greatly benefit from a clear figure.

4) Figure 1E – Is the "shift" = mature adult offspring% – parental% ? If so, I would write that instead of "parent to adult progeny".

5) Figure 2A – The O_2_ consumption measurements relative to the average wildtype basal rate, right? Then say so in the plot.

6) Figure 2B – Why did the authors use very different sample sizes for the three treatments? N=8, 12, and 35 seems a bit varied.

7) “These data show that nutrient sensing via the insulin-signaling pathway regulates sub-organismal proliferation of ∆mtDNA”- A bit of a bold claim. How about, "These data show that nutrient sensing via the insulin-signalling pathway is involved in the regulation of sub-organismal proliferation of ΔmtDNA”?

8) Figure 4F – If higher ΔmtDNA entirely accounts for the higher total copy number, then why not plot the copy number of ΔmtDNA and have this plot as a supplementary?

9) Subsection “Insulin signaling influences sub-organismal proliferation of ∆mtDNA”, third paragraph: Again, rather than "these data show", "these data suggests"… I say this because deletion of *daf-16* partially rescues phenotype.

10) Subsection “Insulin signaling influences sub-organismal proliferation of ∆mtDNA”, fourth paragraph: These few lines need more explanation, especially if you are showing new mutants in the figures. Please expand.

11) Figure 5E: When the y-axis label says "Relative mitochondrial content", what is it relative to?

12) Discussion: I find that the Discussion goes from an overall "big picture" paragraph straight to nutrient-sensing. Can authors discuss Figures 1-3?

Reviewer #3:

In this manuscript, Gitschlag et al. used a heteroplasmic *C. elegans* line carrying both wildtype and *Df5*-deletion mitochondrial genomes to reveal how nutrition status impacts selection on selfish mtDNA at different levels. The authors first characterized and quantified selection on the *Df5* mtDNA at sub-organismal and organismal levels. They then limited nutrition by providing worms with control and restricted diets and explored how that affects the *Df5* propagation at the sub-organismal level. They showed that the *Df5* frequency increase from embryos to adults observed in heteroplasmic worms growing in the normal diet is abolished in worms raised on the restricted diet or worms raised in the normal diet but defective for *daf-2*, the insulin receptor. Deletion of *daf-16* restores the *Df5* increase and proliferation during adult maturation in *daf-2* mutants by restoring the germline development. Interestingly, DAF-16 was shown not to be required for copy-number suppression in response to diet restriction but only required for the *Df5* frequency increase when nutrition is abundant. Finally, they characterized the role of DAF-16 on selection at the organismal level and showed that diet restriction accelerates the decline of *Df5* in *daf-16* mutants, but not in wildtype.

Overall, the manuscript is well written, and the findings are interesting. Below, I included some of my concerns and questions:

1) The sub-organismal selection outcome is a synergistic effect of purifying selection against the *Df5* at the mitochondrial and cell level, and positive selection for the *Df5* due to its replicative advantage. Upon the diet restriction, the author focused on the disappearance of *Df5* increase during adult maturation (Figure 4B and C). However, there is another big change – the disappearance of *Df5* decrease from parents to embryos, which could be due to reduced purifying selection during germline development or increased selfish gain of *Df5* at that stage. This suggests that diet restriction can affect the propagation of *Df5* differently at different developmental stages. I think that it is worth including a discussion on this in the manuscript.

2) I wonder the generality of the nutrition restriction impact on the transmission of selfish mitochondrial genomes described in the *Df5* heteroplasmic line. Is it possible to perform one of the key experiments (e.g. Figure 4C) on a different heteroplasmic line that carries a different type of deletion/selfish mtDNA (e.g. ND5 deletions in natural populations of *C.briggsae*)?

3) Figure 7E and Figure 7—figure supplement 1A showed that non-competing wildtype heteroplasmic flies show a reduced *Df5* genome over generations in control and restricted diet. However, in Figure 2E and Figure 2—figure supplement 1A, such a decline was not observed in the normal diet with wildtype heteroplasmic flies. This indicates that both control and restricted diets affect the cross-generational transmission of *Df5* compared to the normal diet. Please comment on this. It will also be nice if the curve for wildtype flies growing in the normal diet can be included in the same figure for easy comparison.

4) Subsection “Nutrient status governs selection on ∆mtDNA at different levels”, first paragraph: the authors showed that diet restriction accelerated the decline of *Df5* frequency at the level of organismal selection among *daf-16* mutants. Is this acceleration due to much-reduced fertility (e.g. caused by reduced sperm viability or oocyte production) when heteroplasmy is combined with *daf-16* deficiency? For Figure 7C and D, I think it is more important to show how the fraction containing the *Df5* decreases over generations rather than population/individual or population-wide percentage of *Df5*, because the latter two also contain the sub-organismal selection effect, especially for the *daf-16* mutant.

5) I found the presentation of the data not easy to follow. For example, the population/individual calculation was not properly explained in Figure 2E or its legend. I suggest that either the Figure 2—figure supplement 1A and B to be put in Figure 2E, or the figure legend explains what exactly population-wide *Df5* frequency relative to average individual heteroplasmic frequency means. Alternatively, the authors can present only Figure 2—figure supplement 1B in the main figure, which shows how the fraction of heteroplasmic flies decreases over generations. After all, we are looking at competition at the organismal level. Figure 1, Figure 2 and Figure 4E-G legends should contain information on growth conditions including food (e.g. live OP50) and temperature (20 degrees) although such information has been described in the Materials and methods and supplementary information. The presence of the above information in the legend will help readers understand experimental setups without referring to other parts of the paper and make comparisons between data from different figures much easier. The authors also assumed readers to have plenty of pre-knowledge of the *Df5* heteroplasmic line. It will be good to have additional descriptions of the *Df5* heteroplasmic line in the Introduction.

Reviewer #4:

This manuscript reveals a number of interesting and surprising phenomena that influence the relative and absolute abundance of a deletion-bearing mitochondrial genome (ΔmtDNA). It also ties these into a multi-level selection model in an effort to define how organismal and sub-organismal fitnesses are linked. There is much to commend here, and the writing is first-rate. However, I do see a few weakness, which are noted below.

1) The paper's title and overall text focus on the impact of nutritional status, and the signaling pathway that relays this to the cells and tissues, on the replication of the ΔmtDNA genome. However, the huge effect of temperature on even wild-type worms is not given much billing nor explored, beyond serving as an assay to examine the impact of the insulin pathway and nutrition. Is the temperature effect related to the germ line? A re-motivation of the experiments that begins by problematizing the temperature effect would produce a more straightforward paper. For example, the failure to do this makes the following sentence very hard to parse “Together, these data show that DAF-2 signaling inhibits DAF-16 to allow high mtDNA copy number, which permits sub-organismal ∆mtDNA proliferation at the warmer temperature”.

2) Like abundant food, we see that high temp. also promotes ∆mtDNA proliferation, though this requires DAF-2 function. Thus, high temp. acts like abundant food. This is unusual, given that 25°C is somewhat stressful and dauer-promoting. I'm having a hard time reconciling these two results.

3) "We therefore conclude that suppression of germline development by DAF-16 accounts for the reduced mtDNA content in insulin-signaling mutants." We thus have a bit of paradox: growth 25°C doesn't stimulate germline proliferation, yet it does increase ∆mtDNA. So, there may be two independent effects here?

4) The two pairwise comparisons in Figure 6B look qualitatively similar to me. Seems dangerous to build much of an argument around this difference.

---

## [Author Response]

Reviewer #1:The work is interesting, and techniques for quantifying within- and between-host fitness difference between cheating and cooperating mtDNA are nice. The paper does not read smoothly especially at the DAF-16/diet/temperature part. The part of the Price equation is not needed, and seems incorrect/incomplete anyways. Feel free to save the Price equation part which, after a more rigorous analysis, can stand on its own as an independent article.

We appreciate the reviewer’s interest in the work and the techniques, and we do intend to follow this work up by applying these techniques to additional heteroplasmies. As detailed below, we have made extensive revisions to address the concern regarding the writing surrounding DAF-16/diet/temperature results. To address the concerns regarding the Price Equation, we have eliminated this framework from our treatment of multilevel selection, focusing instead on showing how ∆mtDNA frequency itself changes due to selection at each level (within and between host organisms). As per reviewer’s suggestion, we plan to expand on the Price equation analysis and publish it as a separate article, potentially as a Research Advance in *eLife*.

1) The Abstract does not give a clear picture of what the story is. Contents of supplementary figures are rarely given in the main text. Figure legends are also very brief.

We have revised the Abstract for greater clarity and have revised the main text to reference the supplementary information more thoroughly. In addition, the figure legends have been expanded to provide additional relevant information such as diet, temperatures, and ages for all experiments.

2) Figure 1: When cheater mtDNA gets to 80%, it can no longer increase in frequency. Why is that? For the general audience, it is also useful to add a bit more detail on why cheating mtDNA declines from parent to embryo.

We appreciate the reviewer’s interest in these observations regarding the behavior of ∆mtDNA. Although we were initially concerned that addressing these might overly complicate an already detailed study, we agree that these are interesting observations and merit some attention. To this effect, we have addressed these topics by making the following revisions:

1) In the Results section (subsection “An experimental strategy to isolate selection on a selfish mitochondrial genome at different levels”, second paragraph), we now explicitly point out and elaborate on the observation regarding the apparent upper limit of sub-organismal ∆mtDNA proliferation. Additionally, we have expanded the Discussion section (to address this phenomenon of negative frequency-dependent selection (Discussion, third paragraph), whereby the proliferation of ∆mtDNA is attenuated as its sub-organismal frequency increases).

2) We have also expanded the Discussion section (second paragraph), devoting a paragraph to address the phenomenon of purifying selection from parent to embryo.

3) Figure 2C: I am not sure what this means. Does this (decreasing or similar %cheating mtDNA over development) contradict your Figure 1 (increasing %cheating mtDNA over development)?

We appreciate that the way we presented the data in Figure 2C may have caused confusion. We have clarified the presentation of these data in the revised manuscript. In particular, although the frequency of ∆mtDNA increases across larval development, as the reviewer notes, the presence of ∆mtDNA also impedes the rate at which larvae develop. The data in original Figure 2C (now Figure 1F) shows the larval development stage that has been reached, starting from a cohort of age-synchronized embryos. To clarify that the rate of development is negatively associated with ∆mtDNA frequency, we have reversed the axes: the developmental stage reached (vertical axis) is now plotted as a function of ∆mtDNA frequency (horizontal axis). We also reworded the vertical axis label to reflect that the dependent variable is “developmental stage reached,” and we note in the figure legend that these varying-stage larvae began as synchronized embryos, to emphasize that these data reflect an organismal fitness consequence of ∆mtDNA, rather than a sub-organismal dynamic of ∆mtDNA. Accordingly, although ∆mtDNA frequency rises within individuals across development, this rise is not sufficient to conceal the relationship between mutant frequency and the developmental stage that larvae are able to reach within 48 hours.

4) Figure 2E: I find the figure hard to digest. I would like to see three panels plotted in the same figure for comparison: (i) within host advantage of cheating mtDNA (Figure 1D); (ii) between host disadvantage of cheating mtDNA (Figure 2—figure supplement 1B); and (iii) population level mtDNA (Figure 2E). According to Price Equation, (iii) should be a function of (i) and (ii) and variations in cheating mtDNA among offspring embryos.

We appreciate that a main focal point of this paper is the way in which selection acts on ∆mtDNA at two levels, within and between hosts. Accordingly, we have made the following revisions:

1) We combined the ∆mtDNA frequency distribution data (Figure 1C in the original manuscript) and the organismal phenotype data (Figure 2A-C in the original manuscript) into the revised Figure 1. In the revised manuscript, Figure 1 is geared toward establishing ∆mtDNA as a genetic element that undergoes selection both within and between host organisms.

2) We revised Figure 2 to show the quantitative effects of selection at the sub-organismal and organismal levels on ∆mtDNA. To this effect, Figure 2 now contains the intergenerational shift in ∆mtDNA frequency due to sub-organismal heteroplasmy dynamics (panels A and B), a schematic of the organismal selection experiment (panel C), the fraction of ∆mtDNA-containing heteroplasmic hosts per generation (panel D), and the population-wide measurement of ∆mtDNA frequency during the organismal competition experiment (panel E), in accordance with how the reviewer suggested the data be presented.

We thank the reviewer for their suggestions as we think that the new organization of Figures 1 and 2 makes the story much more clear and easier to understand.

5) Price equation: In the Price equation, there are two terms (as suggested in Figure 3—figure supplement 1, although note the typo: () should be ()). The first term of covariance describes how the host fecundity covaries with cooperative mtDNA in the host (and is a function of variations in cheating mtDNA among offspring embryos), and the second term of expectation describes how the cooperative mtDNA is lost to cheating mtDNA within host. Either you have misunderstood Price equation, or your writing is very confusing, especially when you sum up the sub-organismal and organismal covariance terms.

We acknowledge the unfortunate typographical error in the original version of Figure 3—figure supplement 1; the equation should have included the covariance term *E[cov(w_i_,∆z_i_)]* instead of (), corresponding to the version of the Price Equation shown in equation A17 in Price, 1972. Moreover, for the sake of readability, and with the understanding that this work may be of interest to readers outside of evolutionary biology, we have taken the advice of the reviewers and have eliminated the Price Equation framework altogether from our discussion of multilevel selection (and hope to publish it as a separate follow-up article). We therefore have also eliminated the original Figure 3 and its corresponding figure supplements, and renumbered the subsequent figures.

6) Subsection “Insulin signaling influences sub-organismal proliferation of ∆mtDNA”, last paragraph: DAF-16 promotes cheating mtDNA in both rich and poor diet, and not just rich diet, correct? (Figure 6B). In general, the two environmental variables (temperature and diet) plus the genotype variables makes data interpretation and writing (and reading) tricky. Your interpretation has mainly focused on diet despite the apparent importance of temperature (Figure 4F). For example, what happens during restrictive diet and temperature shift? I am not necessarily asking you to do more experiments, but careful writing and illustration is critical when biology gets complex.

Because the effect of warmer temperature on ∆mtDNA proliferation is abolished by the loss of insulin signaling in temperature-sensitive *daf-2* mutants, the variable of temperature provided an opportunity to investigate the role of insulin signaling on the heteroplasmy dynamics of ∆mtDNA. However, we appreciate the suggestion that the effect of temperature itself introduces an additional confounding influence on ∆mtDNA dynamics and warrants discussion. To this effect, we have revised our presentation of the data to address the reviewer’s concerns by making the following changes:

1) The data collected from nematodes incubated at 25°C (the restrictive temperature for the *daf-2*(e1370) allele) is shown in the revised Figure 3 by itself, to compare ∆mtDNA frequency and mtDNA copy number across host nuclear backgrounds under the conditions which inactivate *daf-2*-mediated insulin signaling.

2) The data collected from nematodes incubated at 16°C (the permissive temperature for the *daf-2*(e1370) allele) was moved to Figure 3—figure supplement 1.

3) The Results section has been revised (subsection “∆mtDNA exploits nutrient sensing to proliferate across development”, first paragraph) to emphasize the host genotype effect on ∆mtDNA frequency, while still acknowledging that the difference across nuclear backgrounds requires incubation at the restrictive temperature for *daf-2*(e1370).

4) The putative influence of temperature is now addressed in the fourth paragraph of the revised Discussion section, which is entirely devoted to this phenomenon.

7) Figure 7: This should ideally be the summary figure (data can go to the supplement). You can have schematics for different environments and genotypes. In each schematic, you can include numbers such as selection coefficient of cheating mtDNA from parent to embryo (and variations among embryos), from embryo to Adult (within host advantage, which depends on initial frequency of cheating mtDNA), and at the organismal level (between-host disadvantage). Again, I am not asking you to do more experiments to fill all entries, but putting everything together makes it easier to spot patterns/discrepancies.

We appreciate the reviewer suggestion to use a summary figure to bring greater clarity and understanding of the key results. Having eliminated Figure 3 from the original manuscript, we were able to devote Figure 6 of the revised manuscript to show the effect of diet restriction and *daf-16* genotype on multilevel selection, and use Figure 7 to provide the summary schematic. Additionally, in light of the fact that the role of DAF-16 is complex, we now include schematics in the revised Figures 4-6, which highlight each observed effect of *daf-16* genotype on mtDNA dynamics. To make the overall summary conclusions more apparent and to allow the reader to put everything together, the revised Figure 7 schematic is actually a composite of the schematics from the revised Figures 4-6. Finally, since we no longer discuss multilevel selection using the framework of the Price equation and covariance relationships, we now address selection on ∆mtDNA in terms of changes in mutant frequency itself, rather than converting to selection coefficients.

Reviewer #2:[…]Comment about statistics:1) Are the statistical tests performed in relative or raw datapoints? If they are performed on the relative data, can authors please justify this? I find that by standardising data, differences appear to be bigger and less biologically meaningful.

Several figure panels show data that was normalized to a control condition and/or control genotype (for example the copy number data in the revised Figure 3G, H and Figure 4, and the imaging data in Figure 4). To address the reviewer concern, we performed statistical tests on the raw data, which did not affect the results of the statistical tests. For experiments involving ∆mtDNA frequency, the statistical tests were performed directly on ∆mtDNA frequency, expressed as a percentage of total mtDNA, for each sample.

2) Along these lines, it would be good for the authors to tell the reader in the Results sections about these relative values (how they were calculated, and what they mean). There is a little bit in the figure legends, but this is of great importance for biological conclusions.

We agree with this very important suggestion and have revised the Results section accordingly. This suggestion to explain how we normalize the data is especially important for the data presented in revised Figure 2E. The last paragraph of the subsection “An experimental strategy to isolate selection on a selfish mitochondrial genome at different levels” isdevoted to providing this explanation.

Other comments:1) Subsection “An experimental strategy to isolate selection on a selfish mitochondrial genome at different levels”, second paragraph: I think you mean Figure 1C and D in this sentence instead of Figure 1D and E?

We have separated the data shown in Figure 1C and D into two separate figures, with the Figure 1D data having been moved to Figure 2 for the comparison with organismal selection, as per the recommendation of reviewer 1. We have also revised the main text to make sure the figures are appropriately cited.

2) Figure 1C – It would be very helpful to write on each panel "low", "medium" or "high". I wrote it on my printed copy and it made the figure easier to interpret.

Figure 1C has been revised as per the reviewer’s suggestion.

3) Figure 1D – I understand what you are trying to get to with this figure, but maybe a better way to present it would be using simple boxplots, or mean ± SE (within each of their respected categories). To me it looks a bit messy, and it took me a bit to get my head around it. I'm mainly saying this because this figure conveys a very important message, and the reader would greatly benefit from a clear figure.

As the reviewer suggests, we have replaced the original version of this graph with one that features box-and-whisker plots. But due to this being longitudinal data that features multiple isolated parent-progeny lineages, we also thought it would be appropriate to retain the trajectories in ∆mtDNA frequency, albeit in a more transparent and less distracting format. We reasoned that this is a way to present the data in a less confusing way without sacrificing the longitudinal nature of the data, and is consistent with the visual presentation of data in Figure 1C, D of Wohl et al., *eLife*, 2020 (https://elifesciences.org/articles/52702).

4) Figure 1E – Is the "shift" = mature adult offspring% – parental% ? If so, I would write that instead of "parent to adult progeny".

To address the reviewer concern, this figure (Figure 2B in the revised manuscript) has been revised so that the vertical axis label reads: “Generational shift in % ∆mtDNA (adult progeny – parental).”

5) Figure 2A – The O2 consumption measurements relative to the average wildtype basal rate, right? Then say so in the plot.

The axis in this figure (Figure 1D in the revised manuscript) now reads: “O_2_ consumption rate relative to wildtype basal rate.”

6) Figure 2B – Why did the authors use very different sample sizes for the three treatments? N=8, 12, and 35 seems a bit varied.

The disparity in sample sizes was because we initially collected dozens of heteroplasmic parent lysates for the purpose of correlating ∆mtDNA frequency with fecundity using a linear regression. However, the regression was not significant. We then collapsed the heteroplasmic data points into only two bins (above and below the population mean of 60%) in order to compare their corresponding fertility rates to that of a wildtype control using a Brown-Forsythe and Welch ANOVA. Although we did not find ∆mtDNA frequency to be linearly related to fertility rate, we nevertheless observed the presence of ∆mtDNA to impact fertility compared to wildtype controls. For the sake of consistency and to avoid confusion, we addressed the reviewer concern regarding the very different sample sizes. In particular, we shrank the sample size of the “60-80% ∆mtDNA” bin to N=12 by randomly assigning 12 samples for retaining and 23 samples for discarding. To confirm that this did not affect our statistical analysis, we repeated this three times and performed the statistical analysis on each data set containing a randomly-selected sample of N=12 for the high-∆mtDNA-frequency bin. In each case, the presence of ∆mtDNA at 60-80% frequency continued to correspond to a significantly lower fertility rate than wildtype controls. This information is now provided in the “Fertility” sub-section of Materials and methods.

7) “These data show that nutrient sensing via the insulin-signaling pathway regulates sub-organismal proliferation of ∆mtDNA”- A bit of a bold claim. How about, "These data show that nutrient sensing via the insulin-signalling pathway is involved in the regulation of sub-organismal proliferation of ΔmtDNA.

That statement has been revised in accordance with the reviewer’s suggestion (subsection “∆mtDNA exploits nutrient sensing to proliferate across development”, first paragraph).

8) Figure 4F – If higher ΔmtDNA entirely accounts for the higher total copy number, then why not plot the copy number of ΔmtDNA and have this plot as a supplementary?

The original Figure 4F (Figure 3F in the revised manuscript) has been updated in the following way:

1) The height of the purple data points reflects ∆mtDNA copy number rather than total (wildtype + mutant) copy number, as per the reviewer’s recommendation.

2) The 16°C temperature controls have been moved to the supplement, so that the main figure highlights the comparison of wildtype and mutant copy number across different nuclear backgrounds.

9) Subsection “Insulin signaling influences sub-organismal proliferation of ∆mtDNA”, third paragraph: Again, rather than "these data show", "these data suggests"… I say this because deletion of daf-16 partially rescues phenotype.

That statement has been revised in accordance with the reviewer’s suggestion (subsection “∆mtDNA exploits nutrient sensing to proliferate across development”, third paragraph).

10) Subsection “Insulin signaling influences sub-organismal proliferation of ∆mtDNA”, fourth paragraph: These few lines need more explanation, especially if you are showing new mutants in the figures. Please expand.

We have expanded this part of the Results section to provide a more detailed rationale for the putative roles of mitophagy, fission, and cell death in germline mtDNA content (subsection “∆mtDNA exploits nutrient sensing to proliferate across development”, fourth paragraph).

11) Figure 5E: When the y-axis label says "Relative mitochondrial content", what is it relative to?

These data reflect the quantification of mitochondrial content according to the TOMM-20::mCherry signal; to obtain the relative values, the raw signal across all genotypes was normalized to that of the wildtype nuclear genome. The axis label has been updated to clarify these details.

12) Discussion: I find that the discussion goes from an overall "big picture" paragraph straight to nutrient-sensing. Can authors discuss Figures 1-3?

We appreciate the reviewer pointing out the need to better explain the first third of the figures. The second through fifth paragraphs of the revised Discussion section have been added to more thoroughly address the multilevel selection experiments. In particular, we address the phenomenon of purifying selection from parent to embryo, the negative frequency-dependent sub-organismal selection (whereby the magnitude of ∆mtDNA proliferation was negatively associated with initial frequency), the effect of temperature on sub-organismal ∆mtDNA dynamics, and role of the balancing selection in maintaining ∆mtDNA frequency in a heteroplasmic population.

Reviewer #3:[...]1) The sub-organismal selection outcome is a synergistic effect of purifying selection against the Df5 at the mitochondrial and cell level, and positive selection for the Df5 due to its replicative advantage. Upon the diet restriction, the author focused on the disappearance of Df5 increase during adult maturation (Figure 4B and C). However, there is another big change – the disappearance of Df5 decrease from parents to embryos, which could be due to reduced purifying selection during germline development or increased selfish gain of Df5 at that stage. This suggests that diet restriction can affect the propagation of Df5 differently at different developmental stages. I think that it is worth including a discussion on this in the manuscript.

We have revised the manuscript to address the reviewer’s comment. In particular, the Results section now reports the effect of diet on purifying selection between parent and embryo (subsection “Nutrient availability influences sub-organismal ∆mtDNA dynamics”). In addition, the second paragraph of the revised Discussion section has been added to more thoroughly discuss the relationship between maternal diet and the observed purifying selection against ∆mtDNA between parent and embryo.

2) I wonder the generality of the nutrition restriction impact on the transmission of selfish mitochondrial genomes described in the Df5 heteroplasmic line. Is it possible to perform one of the key experiments (e.g. Figure 4C) on a different heteroplasmic line that carries a different type of deletion/selfish mtDNA (e.g. ND5 deletions in natural populations of C. briggsae)?

The reviewer raises an excellent point regarding the generality of the observations reported in the present study. We chose to focus on *uaDf5* because it is an exceptionally well-characterized selfish heteroplasmic genome with respect to its organismal fitness effects and sub-organismal dynamics. In previous work, we and the Cole Haynes Lab have shown that this mutant genome proliferates, at least in part, by evading host stress-response mechanisms such as mtDNA copy-number control and mitochondrial autophagy, as well as by preferentially benefiting from mtDNA biogenesis. These processes are implicated in heteroplasmy dynamics in other systems, for example in the works of Durham et al. AJHG 2007, Suen et al. PNAS 2010, Kandul et al. Nature Communications 2016, and Chiang et al. Current Biology 2019. Although we therefore expect that the findings of the present study might be generalizable to many other heteroplasmies, we share the reviewer’s sentiment that the generality of these findings should be investigated using other heteroplasmies. To this end, we are equipped with a library of heteroplasmic nematode lines carrying mutations that affect different components of the electron transport chain, which will enable us to identify general principles between the nature of respiratory dysfunction and mutant mitochondrial genome dynamics. Given the scope of this research project, we plan to publish findings from this project in follow-up manuscript, possibly as a Research Advance in *eLife*.

3) Figure 7E and Figure 7—figure supplement 1A showed that non-competing wildtype heteroplasmic flies show a reduced Df5 genome over generations in control and restricted diet. However, in Figure 2E and Figure 2—figure supplement 1A, such a decline was not observed in the normal diet with wildtype heteroplasmic flies. This indicates that both control and restricted diets affect the cross-generational transmission of Df5 compared to the normal diet. Please comment on this. It will also be nice if the curve for wildtype flies growing in the normal diet can be included in the same figure for easy comparison.

The key difference between the non-competing heteroplasmic lines on live food (Figure 2 and Figure 2—figure supplement 1) and the control diet (Figure 6H in the revised manuscript) is that the OP50 bacteria used in the control diet was UV-killed, which we show has a negative effect on ∆mtDNA frequency during development (Figure 3—figure supplement 1C). We have revised the Results section to acknowledge the impact of live versus UV-killed food on the dynamics of ∆mtDNA in the non-competing lines (subsection “Nutrient availability influences sub-organismal ∆mtDNA dynamics”). We have also revised Figure 6H in accordance with the reviewer’s request.

4) Subsection “Nutrient status governs selection on ∆mtDNA at different levels”, first paragraph: the authors showed that diet restriction accelerated the decline of Df5 frequency at the level of organismal selection among daf-16 mutants. Is this acceleration due to much-reduced fertility (e.g. caused by reduced sperm viability or oocyte production) when heteroplasmy is combined with daf-16 deficiency? For Figure 7C and D, I think it is more important to show how the fraction containing the Df5 decreases over generations rather than population/individual or population-wide percentage of Df5, because the latter two also contain the sub-organismal selection effect, especially for the daf-16 mutant.

The reviewer raises an interesting question regarding the nature of the greatly reduced organismal-level fitness when diet restriction is combined with the *daf-16* mutant genotype. Although we do not know the underlying basis of this accelerated selection, we note that FoxO/DAF-16 prolongs organismal survival during nutrient deprivation, as shown in the work of Greer et al., 2007, Kramer et al., 2008, and Hibshman et al., 2018. We speculate that nutrient scarcity and the metabolic dysfunction due to ∆mtDNA represent two sources of energy stress, which additively may compromise host survival and possibly development in the absence of the stress-response functions of *daf-16*.

With respect to the effect of organismal selection on ∆mtDNA, since every individual among the non-competing lines is a carrier of ∆mtDNA, the population-wide frequency of the non-competing lines is equal to the mean heteroplasmic (sub-organismal) frequency. We therefore normalized to sub-organismal frequency by normalizing to the population-wide frequency of the non-competing lines. Moreover, normalizing ∆mtDNA frequency to that of the non-competing lines at each generation sets the overall slope of the non-competing lines to zero. Any significant non-zero slope that remains among the competing lines can be reasonably attributed to the presence of wildtype animals, as this is the only difference between the competing and non-competing lines. Accordingly, normalizing to the frequency of the non-competing lines provides a way to control for sub-organismal selection and isolate the effect of organismal selection on ∆mtDNA frequency. To avoid confusion, we have updated the Results section to explain this method of normalization more clearly (subsection “An experimental strategy to isolate selection on a selfish mitochondrial genome at different levels”, last paragraph). The vertical axis label of Figure 2E has also been updated to clarify that population-wide ∆mtDNA frequency is normalized to the mean sub-organismal frequency. In the competition experiments involving diet and *daf-16*, we used the same approach for controlling for the effect of sub-organismal selection. In addition, we also added the fractions of ∆mtDNA-containing individuals at the conclusion of the competition experiments (generation 8) to the main figures, in accordance with the reviewer’s request.

5) I found the presentation of the data not easy to follow. For example, the population/individual calculation was not properly explained in Figure 2E or its legend. I suggest that either the Figure 2—figure supplement 1A and B to be put in Figure 2E, or the figure legend explains what exactly population-wide Df5 frequency relative to average individual heteroplasmic frequency means. Alternatively, the authors can present only Figure 2—figure supplement 1B in the main figure, which shows how the fraction of heteroplasmic flies decreases over generations. After all, we are looking at competition at the organismal level. Figure 1, Figure 2 and Figure 4E-G legends should contain information on growth conditions including food (e.g. live OP50) and temperature (20 degrees) although such information has been described in the Materials and methods and supplementary information. The presence of the above information in the legend will help readers understand experimental setups without referring to other parts of the paper and make comparisons between data from different figures much easier. The authors also assumed readers to have plenty of pre-knowledge of the Df5 heteroplasmic line. It will be good to have additional descriptions of the Df5 heteroplasmic line in the Introduction.

We acknowledge the complexity of the data. We have taken several steps to address reviewer concerns, including taking their suggestions. Specifically, we have expanded the figure legends to include additional information for every single experiment: diet, temperature, and ages at which animals were used for experiments. Additionally, Figure 2—figure supplement 1B from the original manuscript has been moved to the main figure. The Results section (subsection “An experimental strategy to isolate selection on a selfish mitochondrial genome at different levels”, last paragraph) and the legend accompanying Figure 2E have been revised to more thoroughly explain the rationale and the method for normalizing the population-wide ∆mtDNA frequency to that of the individual/sub-organismal frequency. Finally, the Introduction section has been expanded to provide background information on ∆mtDNA, in accordance with the reviewer’s request (Introduction, fifth paragraph).

Reviewer #4:This manuscript reveals a number of interesting and surprising phenomena that influence the relative and absolute abundance of a deletion-bearing mitochondrial genome (∆mtDNA). It also ties these into a multi-level selection model in an effort to define how organismal and sub-organismal fitnesses are linked. There is much to commend here, and the writing is first-rate. However, I do see a few weakness, which are noted below.1) The paper's title and overall text focus on the impact of nutritional status, and the signaling pathway that relays this to the cells and tissues, on the replication of the ∆mtDNA genome. However, the huge effect of temperature on even wild-type worms is not given much billing nor explored, beyond serving as an assay to examine the impact of the insulin pathway and nutrition. Is the temperature effect related to the germ line? A re-motivation of the experiments that begins by problematizing the temperature effect would produce a more straightforward paper. For example, the failure to do this makes the following sentence very hard to parse “Together, these data show that DAF-2 signaling inhibits DAF-16 to allow high mtDNA copy number, which permits sub-organismal ∆mtDNA proliferation at the warmer temperature”.

We made several revisions to clarify the way in which temperature was used in these experiments, and what we think underlies the effects of temperature on ∆mtDNA frequency. In particular, we now emphasize the comparison across genotype in the main figure, which contains 25°C data – the restrictive temperature that inactivates insulin signaling in the *daf-*2(e1370) mutants – and we moved the data corresponding to the 16°C permissive-temperature controls to the supplement. These revisions are reflected in the first paragraph of the subsection “∆mtDNA exploits nutrient sensing to proliferate across development”. In addition, the fourth paragraph of the revised Discussion section has been added to address the temperature effect.

2) Like abundant food, we see that high temp. also promotes ∆mtDNA proliferation, though this requires DAF-2 function. Thus, high temp. acts like abundant food. This is unusual, given that 25°C is somewhat stressful and dauer-promoting. I'm having a hard time reconciling these two results.

We concur that the combined variables of diet and temperature can make the data difficult to parse, and we thank the reviewer for bringing this to our attention. Although temperature and nutrient deprivation each represent a source of stress, we note that not all stressors are equivalent. In particular, whereas nutrient deprivation impedes ∆mtDNA proliferation, we and others have previously found that the stress caused by the presence of ∆mtDNA itself elicits nuclear-encoded stress responses that promote ∆mtDNA proliferation (Gitschlag et al., 2016 and Lin et al., 2016). We propose that the warmer temperature might stress the animals in some ways that are similar to the presence of ∆mtDNA, which could elicit the amplification of the mutant genome. Accordingly, we have expanded the Discussion section to elaborate on this point (fourth paragraph).

3) "We therefore conclude that suppression of germline development by DAF-16 accounts for the reduced mtDNA content in insulin-signaling mutants." We thus have a bit of paradox: growth 25°C doesn't stimulate germline proliferation, yet it does increase ∆mtDNA. So, there may be two independent effects here?

We agree with the reviewer that the data are indicative of two independent effects. We also agree that our data suggest that temperature increases ∆mtDNA proliferation independent of stimulating germline proliferation. As per the request of other reviewers, who have also noted the importance of addressing the effect of temperature, we have taken two key steps in the revised manuscript. First, we have revised the presentation of the data in Figure 3E-G to focus more on the comparison across genotypes, by moving the data from the permissive 16°C to the supplement. Second, we now devote a paragraph in the Discussion section to discuss the effect on temperature (Discussion, fourth paragraph), and speculate on the potential mechanisms that may underlie these effects. Taken together, we think that the changes we have made should help bring clarity to the presentation of the data, while at the same time acknowledging the effect of temperature.

4) The two pairwise comparisons in Figure 6B look qualitatively similar to me. Seems dangerous to build much of an argument around this difference.

We acknowledge that the difference between wildtype and *daf-16* – with respect to diet-dependent ∆mtDNA proliferation – is modest. However, this difference is significant and reproducible. Importantly, we note that the plots in Figure 5C, D show that across development from embryo to adulthood, increasing copy number coincides with a shift to higher ∆mtDNA frequency; however, this rise is dependent on both food abundance (panel C, compare diets) and also *daf-16* (compare panels C and D, control diet).